# When Are Two Scores Better Than One?
# Investigating Ensembles of Diffusion Models

**Raphaël Razafindralambo**                                      *raphael.razafindralambo@inria.fr*
*Université Côte d'Azur, Inria, CNRS, I3S, Maasai, Nice, France*

**Rémy Sun**                                                                  *remy.sun@inria.fr*
*Université Côte d'Azur, Inria, CNRS, I3S, Maasai, Nice, France*

**Frédéric Precioso**                                    *frederic.precioso@univ-cotedazur.fr*
*Université Côte d'Azur, Inria, CNRS, I3S, Maasai, Nice, France*

**Damien Garreau**                                     *damien.garreau@uni-wuerzburg.de*
*Julius-Maximilians-Universität Würzburg,*
*Institute for Computer Science / CAIDAS, Würzburg, Germany*

**Pierre-Alexandre Mattei**                          *pierre-alexandre.mattei@inria.fr*
*Université Côte d'Azur, Inria, CNRS, LJAD, Maasai, Nice, France*

**Reviewed on OpenReview:** *https: // openreview. net/ forum? id= 4iRx9b0Csu*

## Abstract

Diffusion models now generate high-quality, diverse samples, with an increasing focus on more powerful models. Although ensembling is a well-known way to improve supervised models, its application to unconditional score-based diffusion models remains largely unexplored. In this work we investigate whether it provides tangible benefits for generative modelling. We find that while ensembling the scores generally improves the score-matching loss and model likelihood, it fails to consistently enhance perceptual quality metrics such as FID on image datasets. We confirm this observation across a breadth of aggregation rules using Deep Ensembles, Monte Carlo Dropout, on CIFAR-10 and FFHQ. We attempt to explain this discrepancy by investigating possible explanations, such as the link between score estimation and image quality. We also look into tabular data through random forests, and find that one aggregation strategy outperforms the others. Finally, we provide theoretical insights into the summing of score models, which shed light not only on ensembling but also on several model composition techniques (e.g. guidance). Our Python code is available at `https://github.com/rarazafin/score_diffusion_ensemble`.

## 1 Introduction

Diffusion models have emerged as powerful generative models, demonstrating state-of-the-art performance in applications such as image generation (Dhariwal & Nichol, 2021), text-to-image synthesis (Saharia et al., 2022), video modeling (Ho et al., 2022), graph and molecular design (Liu et al., 2023), and tabular data generation and imputation (Kim et al., 2023; Jolicoeur-Martineau et al., 2024). These models are inspired by diffusion processes which gradually transform complex data distributions into simple ones, and vice versa.

The key idea behind diffusion models is to define a forward process that progressively adds noise to data over multiple steps (Ho et al., 2020; Song et al., 2021b), reaching a "known" distribution, such as Gaussian noise. A reverse process, typically parameterized by a U-Net (Ronneberger et al., 2015), is then trained to denoise the data step by step, reconstructing the original data distribution. Sampling is performed by applying this denoising process iteratively given a model trained to match a score function, starting from the noise.

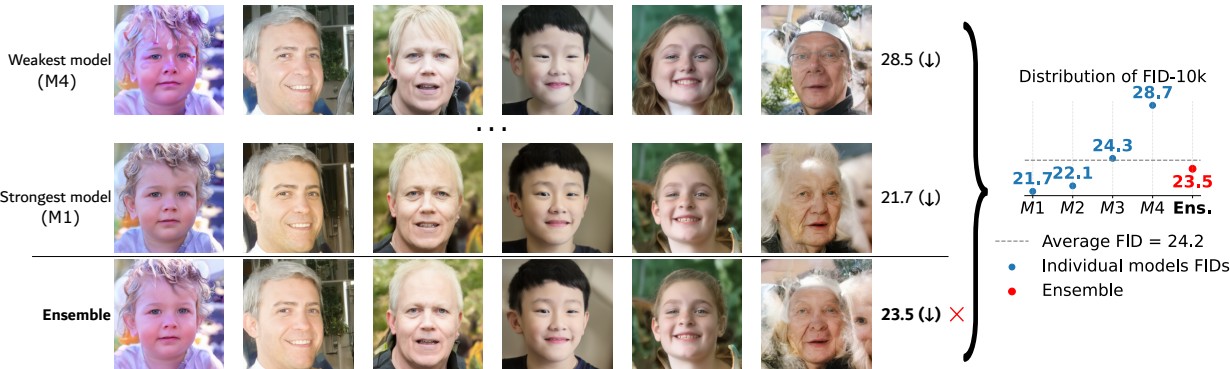

Figure 1: Visual comparison of samples and FID-10k (↓) from two individual models from an ensemble of size $K = 4$ trained on FFHQ-256, the ensemble using the arithmetic mean and DDIM (Section 2.2.2), and the distribution of FID-10k on the ensemble. The initial noise seed is fixed. Ensembling does not clearly improve results: quantitatively, ensembling does not beat the best model. See Section 5.1 for the evolution of two image quality metrics with respect to $K$.

While diffusion models have demonstrated impressive performance in generating high-quality data, the usual strategy to improve their capabilities is either to train deeper neural networks or to extend the training time (Dhariwal & Nichol, 2021; Song et al., 2021b; Rombach et al., 2022). However, this scaling strategy comes with increased computational costs and may not be accessible to all practitioners.

Ensembling (Dietterich, 2000), which combines multiple models' outputs to improve performance, offers an overlooked opportunity to improve diffusion without relying on extensive training of one single model. The idea behind ensembles is that each model contributes different errors and can correct one another's mistakes. These techniques, widely used in statistics and supervised learning, have demonstrated strong empirical results. Ensembling has shown promise for diffusion models as illustrated by the use of Random Forests (an ensemble-based strategy) for unconditional sampling of tabular data (Jolicoeur-Martineau et al., 2024). Forms of ensembling are also used for uncertainty estimation of diffusion models, leveraging entropy over the ensemble's predictions (Jazbec et al., 2025; Kou et al., 2024; De Vita & Belagiannis, 2025), and to improve fidelity when targeting specific distributions (Liu et al., 2022; Du et al., 2023).

Interestingly, attempting to directly apply ensembling to diffusion models does not improve the samples as straightforwardly as one might expect. This is precisely what we observe in the preliminary example of Figure 1 where the outputs of the neural networks, which are the ensemble members, are aggregated using a simple mean of score models. Our observations show that an ensemble of five models does not achieve the highest image quality even though it beats the average one. Yet in supervised learning, neural network ensembles yield noticeable gains as soon as $K = 2$ (Lakshminarayanan et al., 2017) and generally surpass the best individual model (e.g. Hansen & Salamon, 1990, Figure 4; Lee et al., 2015, Table 4).

In this paper we investigate the impact of ensembling on diffusion models across multiple methods. Our contribution is to show that **ensembling yields only marginal, if any, improvements in sample quality**, despite a consistent (but small) reduction in score matching loss. Specifically, we contribute as follows.

1. We look into ensembling for diffusion models (Section 3), using both deep and non-deep base models (Section 3.1), and diverse aggregation strategies (Section 3.2) . We also provide theoretical insights on its effect on the training objective error and contextualize ensembling within other contexts (Section 4).

2. We empirically evaluate multiple ensembling methods and show that, despite improving the training objective as predicted by our theoretical analysis, they bring no significant gains on key metrics such as FID and KID on CIFAR-10 and FFHQ, or the Wasserstein distance on tabular data (Section 5.2.1).

3. We highlight two settings in which ensembling can be helpful. First, the Mixture of Experts provides small benefits when the models are sufficiently complementary, allowing their diversity to be effectively

exploited. Second, in the case of decision-tree aggregation, one method alleviates an underestimation bias issue, leading to improved performance.

4. We investigate factors affecting ensemble performance, including model diversity (Section 5.2.3) and the mismatch between the score matching loss and perceptual quality metrics that could prevent ensembling from showing clear gains (Section 5.3).

## 2 Preliminary on ensembles and diffusion models

### 2.1 Ensemble methods

Ensemble methods trace back to the early days of classical machine learning, where they emerged as a powerful strategy to improve model robustness and accuracy (Dietterich, 2000; Yang et al., 2023). Rather than relying on a single predictor, ensembles aggregate the outputs of multiple models to reduce variance and avoid overfitting. A notable example is the *Random Forest* algorithm (Breiman, 2001), which builds on the principle of *Bagging* (Breiman, 1996) to create a diverse collection of decision trees and combine their predictions. Formally, let $\mathcal{X}$ and $\mathcal{Y}$ denote the input and output spaces. An ensemble is defined as a collection of $K > 1$ predictors, $\left\{ f^{(k)} : \mathcal{X} \to \mathcal{Y}, \quad 1 \leq k \leq K \right\}$, which are combined using an aggregation rule, typically the mean or majority voting. In classification tasks with $L$ classes, the output space $\mathcal{Y}$ corresponds to the $L$-dimensional probability simplex. Over time, ensemble methods have become standard tools in predictive modelling, often outperforming individual models, both in theory (Mattei & Garreau, 2025) and practice (Dietterich, 2000; Grinsztajn et al., 2022). Beyond prediction, ensembles can also be used for generative modelling (Riesselman et al., 2018), clustering (Fleury et al., 2025), or statistical inference (Bach, 2008).

As deep learning became a dominant paradigm in machine learning, ensemble methods were naturally adapted to neural networks. A prominent example is the *Deep Ensemble* (DE), introduced in the 1990s (Hansen & Salamon, 1990) but popularized decades later thanks to its strong empirical results (Lakshminarayanan et al., 2017; Stewart et al., 2023). It involves constructing $K$ models, each sharing the same neural network architecture, which we denote $f_{\boldsymbol{\theta}}^{(k)} \coloneqq f_{\boldsymbol{\theta}_k}$ for $1 \leq k \leq K$ with $\boldsymbol{\theta}_k$ the parameters of the $k$-th model. A key ingredient for its success is the *diversity* between the individual models (Fort et al., 2019). This diversity naturally arises from sources such as random initialization and stochastic gradient descent, even when training on the same dataset. To mitigate the cost of training $K$ separate models, several "train one, get $K$ for free" techniques such as Monte Carlo Dropout (MC Dropout, Srivastava et al., 2014; Gal & Ghahramani, 2016) or Snapshot Ensembles (Huang et al., 2017) have been proposed, which are particularly appealing for diffusion models where training multiple networks is computationally expensive.

Regardless of how the ensemble is constructed, test-time inference requires combining the predictions of the $K$ individual models into a single output. Given a test input $x \in \mathcal{X}$, this is done via an aggregation rule

$$ f(x; K, \boldsymbol{\theta}) = \text{AGG} \left( f_{\boldsymbol{\theta}}^{(1)}(x), f_{\boldsymbol{\theta}}^{(2)}(x), \ldots, f_{\boldsymbol{\theta}}^{(K)}(x) \right). \tag{1} $$

A common approach, particularly in regression and probabilistic settings, is the arithmetic mean, in part because it is the combination that minimizes the average squared loss (Wood et al., 2023).

### 2.2 Diffusion models

In score-based diffusion models, sampling involves learning to estimate the score function. Since we wish to ensemble these estimated scores, we first recall their purpose within the formulation of diffusion models of Song et al. (2021b), based on stochastic differential equations (SDEs).

#### 2.2.1 General framework: score-based SDEs

We introduce three central components that will be used throughout this work: the denoising SDE (Equation (3)), its associated numerical approximation for sampling, and the training loss used to learn the score function (Equation (4)). These components will serve as the basis for explaining how ensembles are trained and how individual models can be combined and evaluated at test time.

Diffusion models aim to generate samples that approximate a given target distribution. Let $q_0(\mathbf{x_0})$ denote this target $d$-dimensional data distribution. Score-based diffusion models (Song & Ermon, 2020; Song et al., 2021a) simulate a process in which data $\mathbf{x}_0 \sim q_0(\mathbf{x}_0)$ is progressively corrupted by noise (the *forward process*) and then denoised to recover the original data distribution (the *backward process*). The forward process is defined by an SDE with continuous time from time 0 to $T$ of the form

$$d\mathbf{x} = \mathbf{f}(\mathbf{x}, t)dt + g(t)d\mathbf{w}, \tag{2}$$

where $\mathbf{f} : \mathbb{R}^d \times [0, T] \to \mathbb{R}^d$ and $g : [0, T] \to [0, +\infty)$ are pre-assigned such that $\mathbf{x}_T$ approximates $\mathcal{N}(\mathbf{0}, \sigma_T^2 \mathbf{I})$, $\mathbf{w}_t \in \mathbb{R}^d$ is a standard Wiener process, and $\mathbf{x}_0 \sim q_0(\mathbf{x}_0)$. Under some regularity conditions, defining $q_t(\mathbf{x})$ as the marginal distribution of $\mathbf{x}_t$ and $\bar{\mathbf{w}}_t$ as a time-reversed Wiener process, a corresponding reverse-time (backward) SDE with the same marginals can be written (Anderson, 1982; Haussmann & Pardoux, 1986)

$$d\mathbf{x} = [\mathbf{f}(\mathbf{x}, t) - g(t)^2 \nabla_\mathbf{x} \log q_t(\mathbf{x})]dt + g(t)d\bar{\mathbf{w}}. \tag{3}$$

To draw samples that approximate $q_0$, we follow the procedure of Song et al. (2021a): start from Gaussian noise $\mathbf{x}_T$ and integrate the backward SDE using typically Euler–Maruyama updates. This involves discretizing the interval $[0, T]$ into $N$ steps $0 = t_0 < \cdots < t_N = T$ and applying the solver from $t_N$ down to $t_0$. Provided the model is sufficiently trained and under additional conditions (De Bortoli, 2022), the final sample $\mathbf{x}_{t_0}$ is distributed according to a density $p_{0,\boldsymbol{\theta}}^{\text{SDE}}$ that approximates $q_0$. Equation (3) also admits a deterministic ODE whose trajectories share the same marginals, allowing the exact computation of the model likelihood via the instantaneous change-of-variables formula (Chen et al., 2018).

The Stein score $\nabla_\mathbf{x} \log q_t(\mathbf{x})$ is the unknown component of Equation (3). Accordingly, score-based diffusion models train to estimate it at each $t$ by minimizing the *Denoising Diffusion Score Matching* (DDSM) objective

$$L_{\text{DDSM}}(\boldsymbol{s_\theta}) := \mathbb{E}_{t \sim [0,T]} \left( \lambda_t \mathbb{E}_{q_{t|0}(\mathbf{x}_t | \mathbf{x}_0) q_0(\mathbf{x}_0)} \| \boldsymbol{s_\theta}(\mathbf{x}_t, t) - \nabla_{\mathbf{x}_t} \log q_{t|0}(\mathbf{x}_t | \mathbf{x}_0) \|^2 \right), \tag{4}$$

where $\lambda_t \equiv \lambda(t) > 0$ is a weighting function, and $\boldsymbol{s_\theta}(\mathbf{x}_t, t)$, generally a time-dependent neural network, aims to predict $\nabla_\mathbf{x} \log q_t(\mathbf{x})$ (Song & Ermon, 2020). Implementations of $\boldsymbol{s_\theta}(\mathbf{x}_t, t)$ include U-Nets (Ronneberger et al., 2015), Vision Transformers (Peebles & Xie, 2023) and Tree-based models (Jolicoeur-Martineau et al., 2024) such as Random Forests (Breiman, 1996) or Gradient Boosted Trees (Friedman et al., 2000).

### 2.2.2 Unifying perspectives on denoising generative processes

The neural network (or any chosen model) estimation of a time-dependent vector field (e.g., Stein score) is the central element of the denoising process for many generative models. While the continuous SDE formulation estimates the score function (Song et al., 2021b) to go to data from noise, other popular generative models such as DDPM (Ho et al., 2020) and DDIM (Song et al., 2020) estimate the noise associated to each step. Flow Matching (Lipman et al., 2023) estimates a velocity field to transport one distribution to another, and Albergo et al. (2025) unify this perspective with diffusion models in a general interpolation framework. Here, we illustrate on DDPMs/DDIMs the unifying framework by showing that they can be treated interchangeably with score-based models when reasoning about the score.

Indeed, DDPMs model the forward and backward process as Gaussian Markov chains and learn to predict noise with a network $\boldsymbol{\epsilon_\theta}(\mathbf{x}_t, t)$. This prediction can be reparameterized into a score estimate as $\boldsymbol{s_\theta}(\mathbf{x}, t) = -\frac{\boldsymbol{\epsilon_\theta}(\mathbf{x}, t)}{\sigma_t}$, where $\sigma_t$ denotes the standard deviation of the marginal distribution at time $t$. This relation allows us to unify DDPMs, DDIMs, and score-based models within a shared ensemble framework.

## 3 How to make an Ensemble of Diffusion models?

The aim of this work is to study the application of ensemble methods to diffusion models. While ensembling in classical machine learning typically involves training multiple models and simply averaging their outputs once, this straightforward strategy does not apply to diffusion models. Their generative process involves a complex iterative procedure, making the aggregation of multiple models non-trivial. We structure this section in two parts: we first explain how to obtain $K$ distinct diffusion models (Section 3.1), and then how to combine their outputs into a single one (Section 3.2). We illustrate the general methodology in Figure 2.

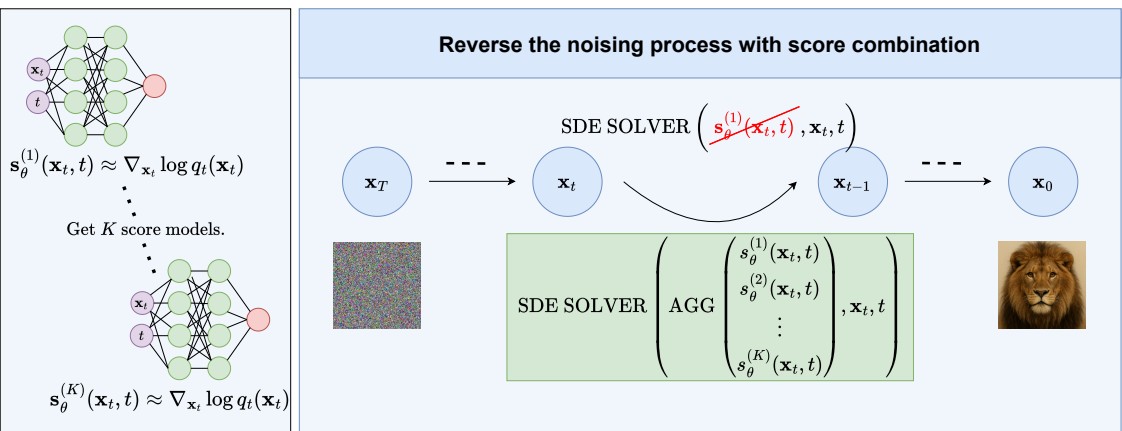

Figure 2: Overview of ensembling within the Diffusion Models framework. We build $K$ score models, for example using Deep Ensemble where each model optimize in parallel the same loss $L_{\text{DDSM}}$. At inference time, we start from noise and generate $\mathbf{x}_0$ using an SDE solver update rule at each step in which we combine the score models using a specific combination rule (e.g. arithmetic mean), instead of using one score model.

## 3.1 Instantiate $K$ score-based predictors

Building on the fact that diffusion models learn a time-dependent score function, we now explore how to build a diverse diffusion model ensemble. We consider both approaches based on neural networks, including independently trained U-Nets, as well as ensemble of decision trees. That said, we will consider DE as the main focus of our empirical study due to its simplicity and widespread use in neural network ensembling.

### 3.1.1 Ensembles of neural networks

We mainly consider the case of neural network ensembles (Hansen & Salamon, 1990). This is typically implemented through Deep Ensemble. To construct a DE of diffusion models, we independently train $K$ U-Nets $\{\boldsymbol{s}_{\boldsymbol{\theta}}^{(k)}\}_{k=1}^{K}$, each initialized with a different random seed. Hence each model is trained separately on the full dataset to optimize the DDSM objective $L_{\text{DDSM}}$ (Equation (4)).

At this stage, the procedure is identical to that of constructing a standard deep ensemble in regression (Lakshminarayanan et al., 2017). A notable point is that the Monte Carlo approximation of the loss exhibits relatively high entropy, due to both the stochastic sampling of timesteps and the injection of Gaussian noise.

Beyond DE, we investigate alternatives techniques to encourage ensemble diversity (discussed in our experiments in Section 5). Indeed Xu et al. (2024) showed that diffusion models with different weight initializations (or architectures) tend to converge to similar functions, despite a natural intuition that stochasticity in the loss computation would lead to diverse solutions. This is explained by the smoothness of the loss landscape, and the result is similar outputs at inference time for the same input noise.

One can use Monte Carlo Dropout (Srivastava et al., 2014; Gal & Ghahramani, 2016; Ashukha et al., 2020) (MC-Dropout) as a lightweight alternative to training $K$ separate U-Nets, which can be computationally prohibitive in terms of training time or memory during inference in our ensemble framework. Indeed MC-Dropout keeps dropout (Srivastava et al., 2014) active at test time on a single model, enabling multiple stochastic forward passes that approximate ensemble behavior without additional training cost.

### 3.1.2 Ensembles of decision trees

Random Forests - which ensemble individual decision trees - can be integrated into diffusion model frameworks to enable efficient training and sampling. Notably, Jolicoeur-Martineau et al. (2024) demonstrated that such tree-based models are particularly effective for tabular data generation, outperforming deep learning–based diffusion methods (Kim et al., 2023; Kotelnikov et al., 2023). The method replaces neural networks

with standard ensemble-based estimators, training a distinct Random Forest at each noise level to approximate the score function. We complement their study by focusing on a new aspect: while they emphasize that XGBoost is the best method to estimate the score (ahead of Random Forest), we analyze the effect of the number of trees forming a Random Forest on performance and study various aggregation schemes.

## 3.2 How to aggregate the $K$ score predictors?

Building on the previous section Section 3.1, where we introduced how to construct an ensemble of $K$ time-dependent models $s_{\boldsymbol{\theta}}^{(1)}(\cdot,\cdot),\dots,s_{\boldsymbol{\theta}}^{(K)}(\cdot,\cdot)$ each trained to estimate the score function at every timestep, we now turn to the question: how can we effectively aggregate these diffusion models? In this section, we introduce novel aggregation schemes for combining the individual score outputs in $\mathbb{R}^d$ from $K$ neural networks at each time step, which we later evaluate in our experiments.

We focus on aggregating diffusion models within a "democratic" ensemble framework, where each model contributes equally to the aggregation process. Furthermore, we assume that the models are exchangeable, meaning their order does not influence the outcome of the aggregation. Based on these principles, we examine several types of ensembles. Unlike traditional ensemble methods in machine learning, where models generate a single prediction for each instance, our framework has to handle outputs at every timestep of the sampling process. Indeed, each diffusion model corresponds to a complete sampling trajectory. Consequently, we consider ensemble approaches that merge score predictions at each timestep. In the following, we adopt the formalism of score-matching, consequently, focus on the neural network that predicts the score. However each of the methods can be applied within other frameworks (Section 2.2.2), and in particular DDPM.

The **Arithmetic mean** of scores, used as a first example in Section 1, is the most natural way to combine the score models at each time step. Let us denote

$$\overline{\boldsymbol{s}}_{\boldsymbol{\theta}}^{(K)}(\mathbf{x},t) := \frac{1}{K}\sum_{k=1}^{K}\boldsymbol{s}_{\boldsymbol{\theta}}^{(k)}(\mathbf{x},t). \tag{5}$$

This is the most common approach for aggregating models to form ensembles, and we provide in Section 3.3 a theoretical analysis that discusses an intuitive though non-rigorous interpretation of this combination rule as a geometric mean of densities. In the same way, we can compute the sum of scores $\sum_{k=1}^{K}\boldsymbol{s}_{\boldsymbol{\theta}}^{(k)}(\mathbf{x},t)$ which also admits a meaningful probabilistic interpretation (see Section 3.3) but likely fails by producing out-of-distribution score outputs at each noise level. The **Geometric mean** and **Median** are alternative schemes, which apply coordinate-wise computations to $K$ vector field outputs as the previous method.

In contrast to averaging schemes, we also consider methods that perform random selection among score models. The **Mixture of experts** strategy choses a unique score model randomly and uniformly prior to sampling, which corresponds to sampling directly from the mixture of densities (see Appendix C.2). In **Alternating sampling**, we exploit the iterative nature of the process by rather randomly selecting a different expert $\boldsymbol{s}^{(k)}(\mathbf{x}_{t_n},t_n)$ at each noise level $t_n$. Finally, in the **Dominant feature** approach, we select at each timestep the component of largest absolute value across all experts, defined for each $i \in \{1,\dots,d\}$ as $[\boldsymbol{s}_{\boldsymbol{\theta}}^{(k^*)}(\mathbf{x},t)]_i$ where $|[\boldsymbol{s}_{\boldsymbol{\theta}}^{(k^*)}(\mathbf{x},t)]_i| = \max_{1\leq k\leq K}|[\boldsymbol{s}_{\boldsymbol{\theta}}^{(k)}(\mathbf{x},t)]_i|$. This strategy selects contributions from the models that have the most to say at each coordinate, with the goal of producing the strongest possible noise and inducing updates of maximal magnitude.

## 3.3 Contextualization of score combinations with other uses

There exists a commonly used connection between step-wise score model aggregation and sampling from a composition of models, given that scores correspond to gradients of log-densities. Model composition usually refers to combining models that have been trained to address different tasks.

A popular example is classier-free guidance (Ho & Salimans, 2022), where scores of different models are summed to guide samples toward a class under a temperature parameter. Specifically, given a class label $\mathbf{y}$, and $\gamma \geq 0$, the goal is to target the distribution with density $\tilde{p}(\mathbf{x}|\mathbf{y}) \propto p(\mathbf{x}|\mathbf{y})p(\mathbf{y}|\mathbf{x})^{1+\gamma} \propto p(\mathbf{x})^{-\gamma}p(\mathbf{x}|\mathbf{y})^{1+\gamma}$. For this distribution, the corresponding score function is straightforward to compute: $\nabla_{\mathbf{x}}\log\widetilde{p}(\mathbf{x}|\mathbf{y}) =$

$\nabla_{\mathbf{x}} \log p(\mathbf{x})^{-\gamma} p(\mathbf{x}|\mathbf{y})^{1+\gamma} = (1 + \gamma)\nabla_{\mathbf{x}} \log p(\mathbf{x}|\mathbf{y}) - \gamma\nabla_{\mathbf{x}} \log p(\mathbf{x})$. Consequently, to sample from the target distribution, the reverse SDE uses the score function given for all $t > 0$ by the approximation

$$\nabla_{\mathbf{x}} \log \widetilde{p}_t(\mathbf{x}|\mathbf{y}) \approx (1 + \gamma)\nabla_{\mathbf{x}} \log p_t(\mathbf{x}|\mathbf{y}) - \gamma\nabla_{\mathbf{x}} \log p_t(\mathbf{x}), \tag{6}$$

where both score functions are estimated using two diffusion models $\mathbf{s}_{\boldsymbol{\theta}}(\mathbf{x}, t \mid \mathbf{y})$ and $\mathbf{s}_{\boldsymbol{\theta}}(\mathbf{x}, t)$. Liu et al. (2022) adopts a generalizing framework of classifier-free guidance to combine multiple features. Alexanderson et al. (2023) apply this approach in another context and refers to it as a Product-of-Experts. In the context of experimental design, Iollo et al. (2025) combine posterior distributions by averaging their scores.

Interestingly, this mirrors our setting: averaging $K > 1$ score models across all $t > 0$ corresponds to combining densities. Specifically, given $K$ probability density functions $p^{(1)}(\mathbf{x}), \dots, p^{(K)}(\mathbf{x})$, the average score $\frac{1}{K}\sum_{k=1}^{K}\nabla_{\mathbf{x}} \log p^{(k)}(\mathbf{x})$ is the score of the normalized geometric mean of the densities, which we define for the remainder of the paper as the *Product-of-Experts* (PoE):

$$\overset{\circ}{p}^{(K)}(\mathbf{x}) := \sqrt[K]{p^{(1)}(\mathbf{x}) \dots p^{(K)}(\mathbf{x})}/Z_K, \quad Z_K = \int \sqrt[K]{p^{(1)}(\mathbf{x}) \dots p^{(K)}(\mathbf{x})}\, d\mathbf{x}. \tag{7}$$

The same terminology has been used to describe generalized geometric means with arbitrary exponents in the context of Gaussian Processes (Liu et al., 2018; Cohen et al., 2020). Applying the same logic as for guidance, ensembling in the backward process for every $t > 0$ would estimate

$$\frac{1}{K}\sum_{k=1}^{K}\nabla_{\mathbf{x}} \log p_t^{(k)}(\mathbf{x}) \approx \nabla_{\mathbf{x}} \log \overset{\circ}{p}_t^{(K)}(\mathbf{x}), \tag{8}$$

where $t > 0$ refers to distributions obtained by noising the data distribution at $t = 0$. Alternatively, summing the scores corresponds to the unnormalized product $\prod_k p^{(k)}(\mathbf{x})$, also referred to as PoE in Hinton (2002).

Some works combine multiple diffusion models in alternative ways. For instance, Balaji et al. (2022); Feng et al. (2023b) combine expert denoisers specialized for different stages of the generative process to better preserve text-conditioning signals. While these approaches do not perform score averaging, they can still be viewed as a form of ensembling, similarly to our Alternating sampling method.

## 4 Does averaging scores make sense? A theoretical insight

In this section, we examine whether averaging scores at each step makes sense from a theoretical perspective. We state that while it provides a simple motivation for ensembling by reducing the $L_{\mathrm{DDSM}}$ objective in expectation (Section 4.1), we have to caution against a natural but misleading interpretation as geometric mean of densities (Section 4.2). We show that in fact diffusion and composing densities in this way do not commute, a relevant point beyond ensembling, as similar compositions arise in guidance-based frameworks.

### 4.1 $K + 1$ models are better than $K$ for score estimation

The following proposition shows that, under mild assumptions, an ensemble of $K$ models $\boldsymbol{s}_{\boldsymbol{\theta}}^{(1)}, \dots, \boldsymbol{s}_{\boldsymbol{\theta}}^{(K)}$ leads to a better expected loss than $K$ models. Such results, mostly based on Jensen's inequality, are classical and well established in machine learning (e.g., regression) for usual convex losses like MSE (McNees, 1992; Breiman, 1996; Mattei & Garreau, 2025). Here we extend them to the score estimation context.

**Proposition 4.1** (Monotonicity for the DDSM loss). *Let $\boldsymbol{s}_{\boldsymbol{\theta}}^{(1)}, \dots, \boldsymbol{s}_{\boldsymbol{\theta}}^{(K)}$ be $K$ score estimators mapping $\mathbb{R}^d \times [0, T]$ to $\mathbb{R}^d$. If for any pair of random variables $(\mathbf{x}_t, t) \in \mathbb{R}^d \times [0, T]$, the outputs $\{\boldsymbol{s}_{\boldsymbol{\theta}}^{(1)}(\mathbf{x}_t, t)\}_{k=1}^{K}$ are identically distributed (i.d.) conditional on $(\mathbf{x}_t, t)$, then*

$$\mathbb{E}\left[L_{DDSM}(\overline{\boldsymbol{s}}_{\boldsymbol{\theta}}^{(K+1)})\right] \leq \mathbb{E}\left[L_{DDSM}(\overline{\boldsymbol{s}}_{\boldsymbol{\theta}}^{(K)})\right]. \tag{9}$$

See Appendix C.1 for a proof. We also discuss in the latter the i.d. assumption of this proposition and provide a similar but more general result on classical score-matching (Hyvärinen, 2005). We also provide results based on KL divergence between probability paths, linking ensembling to improvements in likelihood.

### 4.2 Diffused PoE is not the PoE of diffused distributions

In this section, we show that diffusion and composition do not commute. More precisely, we exhibit a straightforward counter example demonstrating that contrary to natural intuition, adding noise to distributions (e.g., according to a noise scale $\sigma_t$) and then combining them (e.g., geometric mean) does not yield the same result as combining them first and then adding noise. Such result has already been stated in prior work for the product of two densities (Du et al., 2023). Chidambaram et al. (2024) separately provided a counterexample showing that the supports of the resulting distributions generally differ, which consequently implies non-commutativity. Here, we complement these works with an even simpler and more direct one in Section 4.2.1 centered on the non-commutativity. Then, we contextualize the result in Section 4.2.2.

#### 4.2.1 The simple Gaussian counterexample

Let us now formalize this result. We consider the case where the initial distributions are $K$ centered Gaussians with diagonal covariance matrices, i.e., $\mathcal{N}(\mathbf{x}; \mathbf{0}, \alpha_k \mathbf{I})$ for $k = 1, \ldots, K$ with $\alpha_k > 0$. Moreover, we adopt a simplified setting detailed in Appendix C.4. In brief, we focus on a specific instance of the Variance Preserving SDE framework (Song et al., 2021b). To summarize, our counterexample demonstrates that noise and PoE generally do not commute, except when all initial densities are identical, actually reducing PoE to one model.

**Proposition 4.2.** *Let $p_0^{(k)} = \mathcal{N}(\mathbf{0}, \alpha_k \mathbf{I})$ for all $k \leq K$. We denote the PoE of these distributions by $\overset{\circ}{p}_0^{(K)} := \mathrm{PoE}(p_0^{(1)}, \ldots, p_0^{(K)})$. Furthermore, for any density $q_0$, we write $q_t$ the distribution obtained by diffusing it under the forward VP SDE. Then, for any $t > 0$, $\overset{\circ}{p}_t^{(K)} \neq PoE(p_t^{(1)}, \ldots, p_t^{(K)})$ unless $\alpha_1 = \cdots = \alpha_K$.*

#### 4.2.2 Implications for PoE and Diffusion Guidance

Having established that diffusion and PoE do not generally commute, we now turn to discussing the implications in two distinct contexts: the ensemble framework studied in this paper and the guidance setting.

To begin with, Proposition 4.2 straightforwardly shows that Equation (8) is generally **not an equality** for any $t > 0$, as it directly follows from applying the score function. More importantly, and more generally, it demonstrates that step-wise score averaging (resp. summing) does **not** correspond to sampling from a PoE (resp. normalized product of densities). If it were true, such result would be beneficial for two reasons we detail in Appendix C.2 (simplifies theory, eliminates low-probability regions).

Following this, Equation (6) is also **not an equality**, and the guidance method proposed by Ho & Salimans (2022) does not produce the target distribution $\widetilde{p}(\mathbf{x}|\mathbf{y})$, although it approximates it closely. This approach of combining diffusion models should be viewed as a heuristic for approximating the desired distribution. Some previous works have pointed out this subtlety, and proposed methods to target more closely the true distribution using MCMC-based correctors (Du et al., 2023) or the Feynman–Kac equation (Skreta et al., 2025).

These results also extend to classical inverse problems as discussed in Appendix C.5, in which we target a conditional distribution $p(\mathbf{x}|\mathbf{y}) \propto p(\mathbf{x})p(\mathbf{y}|\mathbf{x})$ given an *observed signal* $\mathbf{y}$.

## 5 Experiments

In our experiments, we show that ensembling generally does not lead to consistent improvements in the sample quality of diffusion models. We adopt as our set of practices the study of simple democratic ensembles, that is, ensembles with equal-weight aggregation of independently trained score (or noise) predictors (Section 3.2) sharing the same architecture. We first explore the straightforward strategy of averaging the score predictions of independently trained models at each timestep and find that it fails to enhance visual quality (Section 5.1). We then experiment various ensemble strategies, and aggregation methods ranging from simple arithmetic and geometric means to more elaborate strategies, in an attempt to identify which factors, if any, can mitigate the shortcomings (Section 5.2). We follow the common practice of studying both neural network ensembles and decision tree ensembles (Wood et al., 2023; Theisen et al., 2023). Finally, we seek to explain them by

analyzing the relationship between image quality and the objective the models are actually optimized for within the ensemble (Section 5.3).

We train models on tabular data, on CIFAR-10 (32×32), and on FFHQ (256×256), using different architectures. We measure the FID and KID on 10k samples[1], which are two perceptual metrics, and $L_{\text{DDSM}}$. For a given maximum ensemble size $K_{\max}$, we report confidence intervals for the variability across model subsets when evaluating FID for smaller ensemble sizes (e.g. for $K = 2$ we take all the subsets of size 2 in the whole set of size $K$). For the $L_{\text{DDSM}}$ loss, intervals are computed from 10 runs. In Section 5.3, FID intervals are estimated using bootstrap. KID follows the same procedure as FID. Datasets, model choices, training setups, evaluation methods, and confidence intervals are detailed in Appendix A.

## 5.1 When ensemble averaging fails: insights from image quality metrics

We show that applying simple ensembling using arithmetic mean yields either little or no improvements in image quality despite reducing the score matching loss. In Figures 3a to 3e, we evaluate this approach by progressively increasing the ensemble size $K$ up to $K = 5$ or $K = 10$.

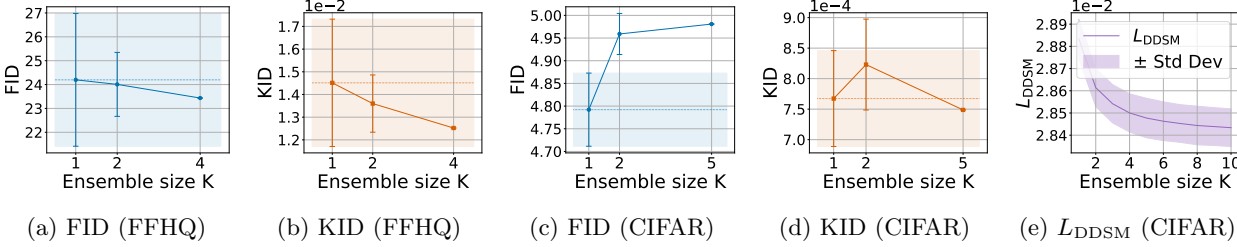

| (a) FID (FFHQ) | (b) KID (FFHQ) | (c) FID (CIFAR) | (d) KID (CIFAR) | (e) $L_{\text{DDSM}}$ (CIFAR) |

Figure 3: Evolution of FID ($\downarrow$) and KID ($\downarrow$) and $L_{\text{DDSM}}$ ($\downarrow$) in function of ensemble size. Samples of the models in Figures 3a and 3b are displayed in Figure 1. $L_{\text{DDSM}}$ is only evaluated on CIFAR-10 since it corresponds to its training objective. See Appendix A for details on each experiment (uncertainty bounds).

This result is surprising since the main heuristic justification of ensembles is that groups are collectively wiser than individuals, as aggregated decisions tend to exhibit reduced variance. Moreover, because the score matching loss, used as the training objective, consistently decreases with larger ensembles, one might expect the models to achieve better approximation of the target distribution and produce improved samples.

Here, however, we find that simple ensembling yields marginal, if any, gains over the average model and never surpasses the best individual model in perceptual metrics such as FID and KID. We stress that surprisingly, as shown on CIFAR-10 in Appendix (Figure 8), ensembling can even perform worse than the weakest individual model.

Figure 10 (Appendix) shows further unsuccessful attempts to use score averaging to compensate for discretization errors with fewer DDIM steps on FFHQ. We also tried applying in Table 7 (Appendix) only during the early sampling steps to test whether averaging is mainly harmful near the data space, but it did not outperform individual models. All these findings suggest that selecting an ensembling method like arithmetic mean that reduces the training loss is insufficient, and that further progress will require going beyond this aggregation strategy.

## 5.2 Investigation on ensemble approaches

We explore a range of ensemble variants to identify promising directions and better understand simple averaging failures. First, we evaluate alternative aggregation strategies (Section 3.2) beyond the arithmetic mean. Next, we evaluate the MC Dropout approach (Section 5.2.2). We also investigate whether encouraging diversity among models can improve performance (Section 5.2.3). Finally we extend our analysis with the non-deep learning approach described in Section 3.1.2 using Forest-VP with Random Forest (Section 5.2.1).

---

[1]The standard is 50k samples. Due to the smaller sample size, FID (often referred to as FID-10k in this context, but we simply refer to it as FID) tends to be overestimated, as it is biased upward.

Table 1: Comparison of different aggregation methods on FID, KID, and $L_{\text{DDSM}}$. The best value for each metric is highlighted in bold. We exclude Alternating Sampling and Mixture of Experts, as $L_{\text{DDSM}}$ reduces to a simple average of individual losses. We focus on aggregations where actual loss reduction is expected. Standard deviations (in parentheses) are reported when applicable; see Appendix A for computation details.

| Deep Ensemble | CIFAR-10 ($32 \times 32$) and $K = 5$ | | | FFHQ ($256 \times 256$) and $K = 4$ | |
| | FID ↓ | KID ↓ | $L_{\text{DDSM}}$ ↓ | FID ↓ | KID ↓ |
| --- | --- | --- | --- | --- | --- |
| Individual models | **4.79** (0.08) | **0.0008** (0.0001) | 0.0284 (0.0005) | 24.20 (5.70) | 0.015 (0.003) |
| Arithmetic mean | 4.98 | 0.0007 | **0.0280** (0.0005) | 23.44 | 0.013 |
| Geometric mean | 4.97 | 0.0008 | **0.0280** (0.0005) | 24.32 | 0.013 |
| Dominant | 7.88 | 0.0041 | 0.0287 (0.0004) | 46.79 | 0.040 |
| Median | 4.98 | 0.0009 | 0.0281 (0.0006) | 23.13 | 0.012 |
| Alternating sampling | 5.05 | 0.0008 | - | 23.07 | 0.012 |
| Mixture of experts | 4.93 | 0.0008 | - | **20.36** | **0.011** |
| Best individual model | 4.65 | 0.0006 | 0.0282 (0.0006) | 21.7 | 0.012 |

### 5.2.1 Comparative study of aggregation techniques

We verify whether the behavior observed by averaging scores holds across all the aggregation methods given in Section 3.2 on both datasets and show that the perceptual improvements are at best marginal (see Table 1).

The four aggregation schemes that combine models at each timestep show no significant differences. On CIFAR-10, these ensembles fail to outperform the average performance of individual models, even though slight improvements in the score matching objective can be observed. On FFHQ, they generally achieve marginal gains over individual models but still do not surpass the best single model. This supports the notion that reducing the training loss alone is insufficient to achieve improvements in perceptual quality.

Among the alternative methods, Mixture of Experts appears most effective, outperforming even the best model on FFHQ. This suggests that randomly selecting a model prior to each sampling run increases diversity across generated samples by leveraging inter-model variability. However, since only one model contributes to each generated image, this approach may not qualify as a true ensemble in traditional sense. Additional results for aggregation schemes, including the sum, are reported in Appendix D.2 as they did not lead to improvements, and are discussed theoretically in Appendices C.1 and D.

### 5.2.2 MC Dropout instead of Deep Ensemble

We also evaluate MC Dropout on CIFAR-10 with up to $K = 20$ models and find that it performs no better—and often worse—than Deep Ensembles (see Table 5 in appendix). FID degrades from 4.83 (no dropout) to 5.85 at best, which corresponds to Arithmetic mean, KID doubles, and $L_{\text{DDSM}}$ slightly increases by 0.0002. We do not measure it on FFHQ since the associated model is not trained with Dropout. The drop in image quality aligns with a deterioration of $L_{\text{DDSM}}$, indicating that MC Dropout's Monte Carlo approximation weakens the network and introduces harmful noise in score estimation. As a result, it not only fails to improve over DE but can even degrade performance on both perceptual metrics and $L_{\text{DDSM}}$.

### 5.2.3 Diversity-promoting strategies

In this section we check if encouraging predictive diversity between models lead to improvements. Indeed ensemble performance is commonly attributed to two factors: (1) the average performance of the individual models, and (2) the diversity in their predictions (Breiman, 1996; 2001; Brown et al., 2005). The idea is that if models make different errors, their combination can reduce overall error through averaging. We focus on the Arithmetic mean and Mixture of experts, as the former serves as our baseline ensemble technique and the latter operates at a different stage of sampling and exhibited the most notable improvements in Table 1.

We first examine the common initialization practice in U-Nets that may hinder diversity in diffusion models (see Figure 4). A near-zero scaling factor $\lambda$ on the terminal weights causes the outputs to be near zero (see details in Appendix G.1). In Table 2, we evaluate how this choice influences the performance of DE, by

Table 2: Comparison of different aggregation methods for $K = 5$ on CIFAR-10 in terms of FID, KID, and $L_{\mathrm{DDSM}}$. We upscaled initialization variance $\lambda$ to one to enhance diversity.

| $\lambda$ upscaled $K = 5$ | CIFAR-10 ($32 \times 32$) | | |
| --- | --- | --- | --- |
| | FID $\downarrow$ | KID $\downarrow$ | $L_{\mathrm{DDSM}} \downarrow$ |
| Individual model DE | **4.79** | **0.0008** | **0.0284** |
| | (0.08) | (0.0001) | (0.0005) |
| Individual model | 5.07 | 0.0010 | 0.0283 |
| | (0.08) | (0.0001) | (0.0004) |
| Arithmetic mean | 5.21 | 0.0011 | 0.0284 |
| | | | (0.0005) |
| Mixture of experts | 5.38 | 0.0010 | - |

Table 3: Ensemble performance for $K = 5$ models trained on overlapping subsets of CIFAR-10 in terms of FID, KID, and $L_{\mathrm{DDSM}}$. Two classes are excluded per model, shifting every two.

| Subsets $K = 5$ | CIFAR-10 ($32 \times 32$) | | |
| --- | --- | --- | --- |
| | FID $\downarrow$ | KID $\downarrow$ | $L_{\mathrm{DDSM}} \downarrow$ |
| Individual model DE | **4.79** | **0.0008** | **0.0284** |
| | (0.08) | (0.0001) | (0.0005) |
| Individual model | 9.65 | 0.0028 | 0.0289 |
| | (0.92) | (0.0006) | (0.0005) |
| Arithmetic mean | 7.13 | 0.0027 | 0.0284 |
| | | | (0.0006) |
| Mixture of experts | 5.86 | 0.0008 | - |

leveling up the scale $\lambda$ to one. We find that this approach does not resolve the lack of improvement observed with ensembles, despite introducing additional diversity in the network's output space (see Appendix G for a broader analysis, including the impact of scaling $\lambda$ on training dynamics and evaluation metrics). This suggests that increasing predictive diversity alone is insufficient to enhance ensemble performance in diffusion models.

Secondly we train ensemble members on different subsets of the dataset as a straightforward way to encourage diversity, and we evaluate this strategy in Tables 3 and 6. For instance, on CIFAR-10, one can assign each model to a specific subset of classes (e.g., two or three classes per model), ensuring that each member specializes in a different part of the data distribution. Combining models specialized on different parts of a dataset is a well-studied area (McAllister et al., 2025; Skreta et al., 2024).

Since each model is specialized in a specific subset of the dataset, one might expect that taking their arithmetic mean for example would merge their respective expertises. However, we find that both aggregation schemes improve image quality but not sufficiently to reach the quality of the models of Table 1. The high FID of Arithmetic mean could result from mass concentration on the intersection of supports, reducing coverage by discarding regions captured by only a subset of models. On Mixture of experts, results are poor in comparison to Table 1 since models are more domain-specific, which could also harm diversity. We analyze the scenario with disjoint supports in appendix (Table 6), revealing an even greater degradation.

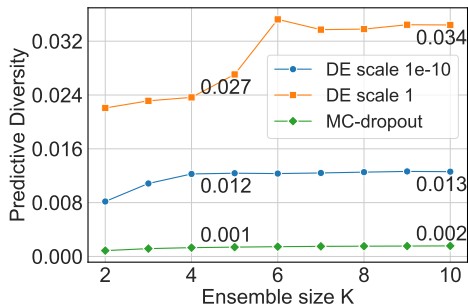

Figure 4: Predictive diversity across three ensemble methods. Increasing $\lambda$ slightly enhances diversity. `DE scale` $\lambda$ denotes Deep Ensemble with models trained on initialization scale $\lambda$. See Appendix G.5 for details on diversity calculation.

Finally we evaluate models trained with varying numbers of iterations (see Figure 5): fewer iterations yield weaker models with more diverse errors which ensembling can possibly compensate effectively. With this in mind, we analyze how ensembling benefits vary with model strength.

We observe that ensembles built from weaker models do yield bigger improvements in FID, while gains diminish and even turn negative as the base models become stronger. Interestingly, training the models longer consistently reduces the loss, highlighting the disconnect between training objective and perceptual quality.

Overall, these experiments show that more diverse ensembles lead to better ensembling effects, especially when base models are weak or specialized. However, this error compensation alone is not sufficient to outperform a single baseline model trained on the full dataset.

### 5.2.4 Random Forests

We evaluate Random Forests as a score predictor and show in Figure 6 that **Dominant components** aggregation yields remarkably stronger performance than other aggregation techniques, that generaly perform poorly. We conduct experiments on various number of trees on the Iris dataset. To assess these settings (details in Appendix A) we evaluate the Wasserstein distance between generated and test data. See Tables 8 and 10 to 13 for more aggregations on larger datasets.

We hypothesize that this particular improvement of Dominant feature strategy compared to others results from a systematic underestimation of the noise magnitude in simple averaging, rather than from ensembling itself.

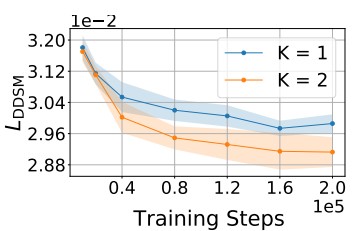
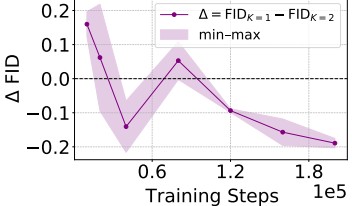

(a) $L_{\text{DDSM}}$ vs Training Steps

(b) $\Delta$FID vs Training Steps

Figure 5: Comparison of DE ($K$=2) and individual model ($K$=1) in terms of training loss and FID, plotted against the number of training iterations. The difference stays positive for $L_{\text{DDSM}}$ (a), while for FID (b), for which we directly plot it, the $\Delta$ varies from positive to negative. See Appendix A for details on the CI.

To check this, for each perturbed data $\boldsymbol{X}_t \in \mathbb{R}^{n \times d}$, we fit a Random Forest regressor to predict the noise that transforms the original data point $\boldsymbol{X}_0 \in \mathbb{R}^{n \times d}$ into $\boldsymbol{X}_t$. This noise is in fact systematically underestimated during generation, as shown in Figure 11. Arithmetic mean exhibits a lower overall standard deviation for all $t > 0$, supporting the hypothesis of a statistical bias, namely that it fails to capture the true noise magnitude.

### 5.3 Correlation between Score Matching and FID

As we have observed across previous experiments, the $L_{\text{DDSM}}$ function used during training and perceptual metrics remain poorly aligned in ensemble evaluation. As shown in Section 5.1, score matching can benefit from increasing $K$ when perceptual metrics do not.

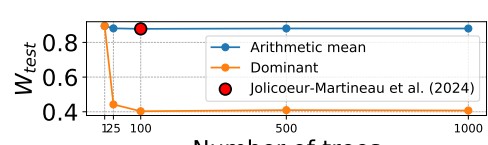

Figure 6: Wasserstein distance $W_{\text{test}}$ ($\downarrow$) for Arithmetic mean and Dominant feature methods as a function of the number of trees $K$. We perform 3 train/test splits for each experience, and report the mean.

This suggests that ensembling might be effective to some extent, as it enhances score estimation which is the very objective the models are trained to optimize.

In this section, we highlight the gap between these two metrics. We also show that the gap in behavior between score-based metrics and FID is particularly noticeable when noise is introduced into the score function.

### 5.3.1 Score matching and FID during training

We show in Figure 7a that already over the training course, the behavior of the optimized objective differs from the one of the perceptual metric of interest. While FID keeps improving steadily, the validation score matching loss drops sharply early on and then stagnates. This shows that the score matching loss alone is not a sufficient indicator on the performance of the model in generation. These results are obtained by evaluating both metrics on a single score model across checkpoints up to 200k iterations.

### 5.3.2 Noising the score and its effect on FID

We show in Figure 7b that perturbing the score in the regime where ensemble improvements typically affect the loss leads to significant and chaotic variations in FID At each timestep, we apply perturbations to the score model by adding $\tau \|\boldsymbol{s_\theta}\|_2 \mathbf{z}$ to its output, where $\tau \in [0, 1]$ is a fixed noise level, and $\mathbf{z}$ is uniformly distributed on the unit sphere. These results highlight that FID does not evolve in a predictable manner in comparison to the score, showing that intuitive interpretations of FID variations can be misleading.

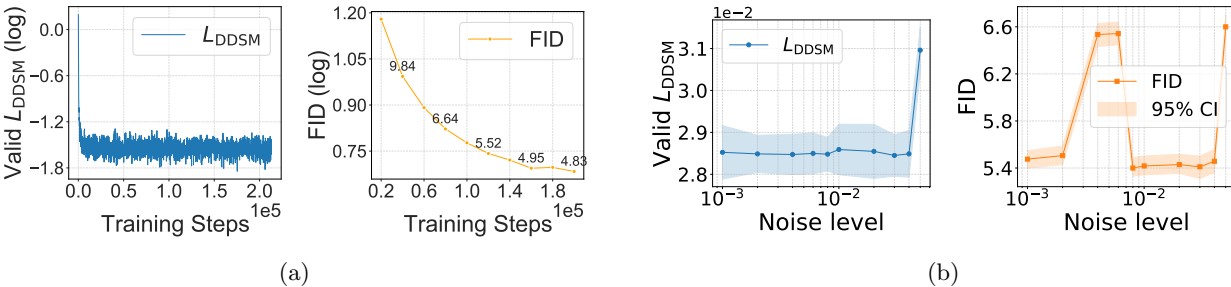

Figure 7: Evolution of score matching loss and FID (in log scale) during training (a), and sensitivity of FID and $L_{\text{DDSM}}$ to noise levels $\tau$ (in log scale on the figure) in the score function (b). In both cases, trends suggest that the FID evolution cannot be reliably predicted from the score dynamics.

The disconnect between what diffusion models are trained to optimize (score matching) and how they are typically evaluated explains why ensembling struggles to diminish image quality metrics. This is further reflected in the decreasing behavior of the likelihood (see Figure 9), which has a close mathematical link (up to first or second-order derivatives (Song et al., 2021a; Lu et al., 2022)) with the score estimation. Echoing our observation in this section, Theis et al. (2016) and van den Oord & Dambre (2015) pointed out a generally accepted point in the generative model community, that high likelihood does not necessarily translate into better sample quality. Our experiments are another illustration of this point.

## 6 Conclusion

We investigated ensemble techniques for diffusion models on unconditional generation, starting with straight-forward aggregation rules then extending to more heuristic or diversity-promoting (but still simple) strategies. Averaging a Deep Ensemble, the most natural approach, failed to boost perceptual metrics such as FID and KID. Motivated by this, we tried alternative aggregation schemes, dropout-based ensembles, subset training, and initialization tweaks; none consistently closed the gap. We also evaluated Random Forests for the tabular data generation framework, and found that using the component with the highest amplitude across the trees improves sample quality based on Wasserstein distance.

The main observation is that ensembling generally yields only marginal gains in FID. The only consistent improvement observed with Deep Ensembles concerns score matching, which is largely a mechanical effect resulting from Jensen's inequality and the models' exchangeability. Nevertheless, we observe a few notable exceptions where specific forms of ensembling can yield more perceptible benefits in particular settings: the Mixture of Experts performs better when the models are truly complementary, suggesting that the diversity captured by independently trained models is valuable. However, averaging their scores tends to blur this diversity, thereby degrading perceptual quality. Overall, these observations reinforce the main disconnect: better score matching does not necessarily translate into higher perceptual quality, which explains the limited impact of standard ensembling.

Regarding the benefit–cost trade-off, our experiments suggest that standard ensembling offers limited returns relative to its computational expense. While a couple of approaches can provide more improvements, these gains remain rare, context-dependent. In addition, the substantial increase in training and sampling cost rarely justifies their use in diffusion models. This stands in contrast with supervised learning, where ensembling is known to provide consistent and significant performance gains (Hansen & Salamon, 1990; Lee et al., 2015) that clearly outweigh its computational overhead.

We showed that the mean of the scores at each step does not sample the PoE. While Feynman–Kac (Skreta et al., 2025) or MCMC-based (Du et al., 2023) correctors could in principle adjust the sampling process to better target the PoE, we deliberately omitted them to restrict the work to baseline ensembling. Moreover, guidance-based approaches already perform well without the additional theoretical rigor introduced by such corrections. Nonetheless, future work could further explore this direction.

We voluntarily restricted our analysis to "democratic" ensembles, where all models contribute equally and independently. Future work aiming to improve diffusion models through ensembling could explore more advanced strategies, such as train one, get $K$ for free or other low-cost ensemble methods (Ramé et al., 2021; Huang et al., 2017; Ha et al., 2017; Havasi et al., 2021; Wasay et al., 2020). Another promising direction would be to relax the equal or independent constraint, for instance through weighting or boosting.

**Acknowledgments**

This work was supported by the French government, through the 3IA Côte d'Azur Investments in the Future project managed by the National Research Agency (ANR) with the reference numbers ANR-19-P3IA-0002 and ANR-23-IACL-0001. DG also acknowledges the support of ANR through project NIM-ML (ANR-21-CE23-0005-01). Parts of this work were done as DG was employed at Université Côte d'Azur.

We thank the reviewers for their constructive feedback, which helped improve the quality and clarity of this paper. We are also grateful to Florence Forbes for carefully pointing out an error in an earlier version of the paper.

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

This appendix supplements our main contributions with additional insights, methodological clarifications, and extended empirical results:

- Appendix A details the experimental settings used throughout the study.

- Appendix B outlines how to assess step-wise aggregation strategies beyond sampling, including evaluations of $L_{\mathrm{DDSM}}$ and model likelihood.

- Appendix C elaborates on the theoretical motivations, formulation, and implications of the proposed aggregation mechanisms. In particular, we prove Proposition 4.1 and Proposition 4.2.

- Appendix D presents exploratory yet ineffective alternatives for combining diffusion models.

- Appendix E and Appendix F provide additional plots and tables supporting our experimental findings in Section 5.

- Appendix G examines and contextualizes the popular yet understudied practice of downscaling final-layer weights at initialization in diffusion models.

- Appendix H offers a short commentary on the limitations of FID and the caution required when interpreting it.

## A    Experimental details

We describe in detail both our deep and non-deep approaches.

### A.1    Ensemble of U-Nets

We train our models on low-resolution images using CIFAR-10 (32×32) (Krizhevsky, 2009). To assess the scalability to higher-resolution data, we also conduct experiments on FFHQ (256×256) (Karras, 2019).

For experiments on CIFAR-10, our neural network follows the DDPM++ continuous architecture as proposed by Song et al. (2021b). We use the VP-SDE formulation discretized with 1000 diffusion steps, and a predictor-only sampling procedure by adopting the Euler-Maruyama scheme. Training parameters (learning rate, batch size, etc.) and diffusion parameters $(\bar{\beta}_{\min}, \bar{\beta}_{\max})$ are identical to those in the original paper; in particular, we set the dropout rate to 0.1. Each model is trained for 200k iterations. For FFHQ-256, we use the ADM architecture from OpenAI's official repository, which includes several improvements over the previous U-Net design (Dhariwal & Nichol, 2021). The model is trained using the DDPM objective. We follow the repository's recommendation for the learning rate (1e-4), while the batch size (4) and total number of training iterations (10M) are chosen based on our computational resources. No dropout is used in this setup. At sampling time, we use the DDIM sampler with 100 steps with an entropy level $\eta = 0.5$ (see Song et al. (2020) for a definition) by default. Unless otherwise specified, these settings are consistently applied accross all experiments presented in this work.

We evaluate the perceptual quality of generated images using the Fréchet Inception Distance (FID) (Heusel et al., 2017) and Kernel Inception Distance (KID) (Bińkowski et al., 2018). On the one hand, FID measures the distance between the feature distributions of real and generated images using activations from a pre-trained Inception network (Szegedy et al., 2016), and assuming the features are Gaussian-distributed. On the other hand KID computes the Maximum Mean Discrepancy (MMD) with a polynomial kernel. All metrics are computed on 10k samples instead of the standard 50k. Due to the smaller sample size, FID (often referred to as FID-10k in this context, but we simply refer to it as FID) tends to be overestimated, as it is biased upward.

### A.2    Random Forests

We conduct experiments on Forest-VP by varying the number of trees, denoted as $K$ (corresponding to `n_estimators` in the scikit-learn implementation). The values considered are $\{1, 10, 25, 50, 100, 500, 1000\}$,

where $K = 100$ is the default setting used by Jolicoeur-Martineau et al. (2024). Moreover, we consider various aggregation methods between the trees.

We adopt the same training and sampling procedure as the reference model in the original paper (Jolicoeur-Martineau et al., 2024, Table 6). In particular, we set `max_depth = 7` following the cited work[2], which is sufficient for small datasets such as *Iris*.

To ensure consistency with Jolicoeur-Martineau et al. (2024), we evaluate Random Forest using the Wasserstein distance between the generated and test data, coverage (Naeem et al., 2020), and efficiency (Xu et al., 2019), with results reported in Tables 8, 12 and 13. Coverage assesses the diversity of generated samples relative to the test set. Efficiency, is measured as the average F1 score obtained from training multiple non-deep machine learning models for classification or regression on generated data and evaluated on the test set.

Alongside the Iris dataset ($n = 150$, $d = 4$), we report additional results on larger datasets ($n > 1000$) in Appendix F.1.

### A.3 Confidence intervals

In this section, we detail how confidence intervals are computed for the different experiments presented in the paper. We distinguish three cases: 1) FID/KID evaluation on ensembles, 2) $L_{\mathrm{DDSM}}$ evaluation on ensembles, and 3) FID/KID evaluation in Section 5.3.

#### A.3.1 Standard deviations on ensembles

For FID and KID evaluations on ensembles of size $K = 5$ in Figures 3a to 3d and 5b and tables 1 to 3 and similar reports in Appendix, variability is assessed across different model subsets and not through repeated metric computation (due to the computational cost of FID on ensembles). For individual models, the interval reflects the variability across the individual models taken separately. For subsets such as 2 models, we report the variability obtained by evaluating all possible pairs among the five available models (i.e., $\binom{5}{2}$ combinations). For both configurations, we compute the mean and standard deviation of these measurements. For the full ensemble ($K = 5$), the metric is computed only once since all models are included.

In the case of Figure 5b, where $K = 2$ corresponds to the full ensemble, we use the minimum and maximum values to represent the uncertainty on $K = 1$, as this choice is more appropriate than taking the standard deviation.

For the $L_{\mathrm{DDSM}}$ evaluation (Figures 3e and 7a and table 1 and similar reports in Appendix), we are able to measure the metric over multiple model combinations for each ensemble size, and repeat each measurement up to 10 times thanks to its low computational cost.

#### A.3.2 Intervals for a fixed model w/ bootstrap

In Figure 7b, since we rely on a single fixed model and require a precise estimation of the trend of these perceptual metrics, we assess variability across measurements using the bootstrap method (Hastie et al., 2009, Section 7.11), which offers a cost-effective alternative to repeated evaluations. The bootstrap method estimates variability by repeatedly resampling (100 times in our case), with replacement, from the original set of measurements. Each resample is used to recompute the metric, producing an empirical distribution of estimates. The confidence interval can be estimated for example from the standard deviation of the bootstrap distribution or by taking its 2.5th and 97.5th percentiles for a 95% interval, which is what we do. FID is a biased estimator, as fewer samples tend to increase its value. Standard percentile bootstrapping can therefore introduce additional bias due to resampling with replacement. To ensure that bootstrap samples contain about $10\,000$ *unique* observations (matching FID-10k), we adjust the sample size accordingly as follows.

---

[2] https://github.com/SamsungSAILMontreal/ForestDiffusion

In a bootstrap resample of size $n$, each observation has a probability of not being selected given by

$$\left(1 - \frac{1}{n}\right)^n \approx e^{-1} \approx 0.368, \tag{10}$$

which means that about 63.2% of observations are unique on average. To obtain 10 000 unique samples, we solve

$$0.632\, n = 10\,000, \tag{11}$$

which gives

$$n \approx 15\,823. \tag{12}$$

Thus, using bootstrap samples of size 15 823 ensures roughly 10 000 distinct observations per resample, aligning the evaluation with FID-10k.

## B   How to evaluate score matching and model likelihood on arithmetic mean?

Given a trained model $\boldsymbol{\theta}$, the procedures for computing $L_{\text{DDSM}}(\boldsymbol{\theta})$ and $p_{0,\boldsymbol{\theta}}^{\text{ODE}}(\mathbf{x}(0))$ are well established, as explained in Section 2.2.1 and by Song et al. (2021b). Thus, if $\{\boldsymbol{\theta}_k\}_{k=1}^K$ denotes a collection of $K$ set of parameters optimized using DE, we essentially evaluate the metrics for model averaging by plugging in $\overline{\boldsymbol{s}}_{\boldsymbol{\theta}}^{(K)}(\mathbf{x}, t) := \frac{1}{K} \sum_{k=1}^K \boldsymbol{s}_{\boldsymbol{\theta}}^{(k)}(\mathbf{x}, t)$ into their formula. In the following we provide details for implementation of this particular case, but it can be extended to any step-wise aggregation.

### B.1   Score matching loss of arithmetic mean

Let $\mathbf{x}_0 \in \mathcal{D}_{\text{valid}}$. We sample $t \in [0, T]$, and $\mathbf{z} \sim \mathcal{N}(\mathbf{0}, \boldsymbol{I})$ to compute $\mathbf{x}_t \sim q_{t|0}(\mathbf{x}_t | \mathbf{x}_0)$ using $\mathbf{x}_0$ and the explicit formula of the forward transition (see Song & Ermon, 2020, Appendix B). The only difference is that we replace the individual instance of a model by an average. For example, if we consider an SDE with zero drift and noise schedule $\sigma : [0, T] \to \mathbb{R}_{>0}$, $L_{\text{DDSM}}$ is computed using $\|\overline{\boldsymbol{s}}_{\boldsymbol{\theta}}^{(K)}(\mathbf{x}_t, t) + \frac{\mathbf{z}}{\sigma(t)}\|^2$ which is averaged using MC over the distribution of $\mathbf{x}_0$ and the timesteps $t$.

### B.2   Model likelihood of arithmetic mean

Before discussing ensembles, let us first present a short review of likelihood calculation for diffusion models. Associated to Equation (3), there exists a corresponding deterministic process following an ODE and whose trajectories $\{\mathbf{x}_t\}_{t=0}^T$ share the same marginal densities $\{q_t(\mathbf{x})\}_{t=0}^T$ as the SDE:

$$\frac{\mathrm{d}\mathbf{x}}{\mathrm{d}t} = \tilde{\mathbf{f}}(\mathbf{x}, t) := \left[\mathbf{f}(\mathbf{x}, t) - \frac{1}{2}g(t)^2 \nabla_{\mathbf{x}} \log q_t(\mathbf{x})\right]. \tag{13}$$

In this case, we denote $p_{t,\boldsymbol{\theta}}^{\text{ODE}}(\mathbf{x}_t)$ the probability at each time $t$ induced by the parameterized ODE. Under this framework, we have a particular case of Neural ODE/Continuous Normalizing Flow (Chen et al., 2018). Consequently, the likelihood $p_{0,\boldsymbol{\theta}}^{\text{ODE}}(\mathbf{x}_0)$ can be explicitly derived by using the instantaneous change-of-variables formula that connects the probability of $p_{0,\boldsymbol{\theta}}^{\text{ODE}}(\mathbf{x}_0)$ and $p_{T,\boldsymbol{\theta}}^{\text{ODE}}(\mathbf{x}_0)$, given by

$$p_{0,\boldsymbol{\theta}}^{\text{ODE}}(\mathbf{x}(0)) = e^{\int_0^T \nabla_{\mathbf{x}} \cdot \tilde{\mathbf{f}}(\mathbf{x}(t),t)\, dt}\, p_{T,\boldsymbol{\theta}}^{\text{ODE}}(\mathbf{x}(T)), \tag{14}$$

where $(\nabla_{\mathbf{x}} \cdot)$ denotes the divergence function (trace of Jacobian w.r.t $\mathbf{x}$). For the SDE counterpart, Song et al. (2021a) established that $\log p_0^{\text{SDE}}(\mathbf{x}(0))$ admits a tractable lower bound, while Albergo et al. (2025) newly derived a closed-form expression for it.

Now we explain how to compute this for an arithmetic mean of $K$ models. We denote by $\{\mathbf{x}_{\boldsymbol{\theta}}(t)\}_{t=0}^T$ the trajectory of the ODE associated with the score model $\overline{\boldsymbol{s}}_{\boldsymbol{\theta}}^{(K)}(\mathbf{x}, t)$. Although $\{\mathbf{x}_{\boldsymbol{\theta}}(t)\}_{t=0}^T$ depends on the full set of models $\boldsymbol{\theta}_1, \ldots, \boldsymbol{\theta}_K$, we refer to it using the shorthand $\boldsymbol{\theta}$ for simplicity, following the notation used for the

average score function. The log-likelihood can be exactly calculated by numerically solving the concatenated ODEs backward from $T$ to $0$, after initialization with $\mathbf{x}_{\boldsymbol{\theta}}(0) \sim q_0$ (`solve_ivp` from `scipy.integrate`),

$$\frac{d}{dt} \begin{bmatrix} \mathbf{x}_{\boldsymbol{\theta}}(t) \\ \log p_{t,\boldsymbol{\theta}}^{\mathrm{ODE}}(\mathbf{x}_{\boldsymbol{\theta}}(t)) \end{bmatrix} \tag{15}$$

$$= \begin{bmatrix} f(\mathbf{x}_{\boldsymbol{\theta}}(t), t) - \frac{1}{2} g^2(t) \overline{\boldsymbol{s}}_{\boldsymbol{\theta}}^{(K)}(\mathbf{x}_{\boldsymbol{\theta}}(t), t) \\ \frac{1}{2} g^2(t) \nabla_{\mathbf{x}} \cdot \overline{\boldsymbol{s}}_{\boldsymbol{\theta}}^{(K)}(\mathbf{x}_{\boldsymbol{\theta}}(t), t) - \nabla_{\mathbf{x}} \cdot \mathbf{f}(\mathbf{x}_{\boldsymbol{\theta}}(t), t) \end{bmatrix}. \tag{16}$$

The divergence $\nabla_{\mathbf{x}} \cdot (.)$ is approximated using the Hutchinson Monte-Carlo based estimator of the Trace (Hutchinson, 1989), motivated by the identity

$$Tr(A) = \mathbb{E}_{\mathbf{z} \sim \mathcal{N}(\mathbf{0}, \boldsymbol{I})}[\mathbf{z}^T A \mathbf{z}] \implies \nabla_{\mathbf{x}} \cdot \overline{\boldsymbol{s}}_{\boldsymbol{\theta}}^{(K)}(\mathbf{x}_{\boldsymbol{\theta}}(t), t) \approx \frac{1}{M} \sum_{m=1}^{M} \mathbf{z}_m \, \nabla_{\mathbf{x}} \overline{\boldsymbol{s}}_{\boldsymbol{\theta}}^{(K)}(\mathbf{x}_{\boldsymbol{\theta}}(t), t) \, \mathbf{z}_m^T \tag{17}$$

where $\mathbf{z}_1, \ldots, \mathbf{z}_M \sim \mathcal{N}(\mathbf{0}, \boldsymbol{I})$. Following Song et al. (2021b), we set $M = 1$. To correctly compute the NLL, dequantization techniques are employed; see Theis et al. (2016) for details and justification.

## C  Theoretical insights on model aggregations

In this section, we complement our theoretical findings by providing proofs and additional implications of the results. Appendix C.1 addresses the denoising score matching loss whose associated result is Proposition 4.1; Appendix C.2 expands on Section 3.3, notably by including Mixture of Experts in the analysis; Appendix C.3 develops the misconception raised in Section 4.2; and Appendix C.4 provides the detailed framework and proof of the main result stated in Proposition 4.2, followed by a discussion of the contexts implied by this result in Appendix C.5.

### C.1  Arithmetic mean and model performance

In this section, we provide the proof of Proposition 4.1, which demonstrates the benefit of ensembling on $L_{\mathrm{DDSM}}$. We first establish a preliminary result showing that an ensemble of $K$ models performs better than a single model, before turning to the main result that $K+1$ models outperform $K$. We also derive a short consequence concerning the KL divergence between probability density paths.

**K+1 models are better than K for score estimation.**  The training objective of diffusion models is to minimize the denoising score matching loss in Equation (4), which computes the mean squared error between the estimated score and the true conditional score given clean images, averaged over all timesteps. This formulation has the advantage of reducing the inherently difficult generative modeling task to a series of simpler supervised regression problems. Given any positive time-dependent weighting function $\lambda(\cdot)$ and model $\boldsymbol{s}_{\boldsymbol{\theta}}(\cdot)$, we define an equivalent definition to Equation (4),

$$\mathcal{L}_{\mathrm{DDSM}}(\boldsymbol{s}_{\boldsymbol{\theta}}, \lambda(\cdot)) := \frac{1}{2} \int_0^T \mathbb{E}_{p_0(\mathbf{x}_0) p_{t|0}(\mathbf{x}_t | \mathbf{x}_0)} \left[ \lambda(t) \| \boldsymbol{s}_{\boldsymbol{\theta}}(\mathbf{x}_t, t) - \nabla_{\mathbf{x}_t} \log q_{t|0}(\mathbf{x}_t | \mathbf{x}_0) \|_2^2 \right]. \tag{18}$$

In the following we demonstrate that when models from an ensemble of size $K$ are similar, adding one model improves the accuracy of our score estimation in average.

**Proposition C.1.** *Let $\boldsymbol{s}_{\boldsymbol{\theta}}^{(1)}, \ldots, \boldsymbol{s}_{\boldsymbol{\theta}}^{(K)}$ be $K$ score estimators mapping $\mathbb{R}^d \times [0, T]$ to $\mathbb{R}^d$. Then, the DDSM loss satisfies*

$$\mathcal{L}_{DDSM}(\overline{\boldsymbol{s}}_{\boldsymbol{\theta}}^{(K)}, \lambda(\cdot)) \leq \frac{1}{K} \sum_{j=1}^{K} \mathcal{L}_{DDSM} \left( \frac{1}{K-1} \sum_{k \neq j} \boldsymbol{s}_{\boldsymbol{\theta}}^{(k)}, \ \lambda(\cdot) \right) \tag{19}$$

*Moreover, if for any $(\mathbf{x}_t, t) \in \mathbb{R}^d \times [0, T]$, the outputs $\boldsymbol{s}_{\boldsymbol{\theta}}^{(1)}(\mathbf{x}_t, t), \ldots, \boldsymbol{s}_{\boldsymbol{\theta}}^{(K)}(\mathbf{x}_t, t)$ are identically distributed, then,*

$$\mathbb{E} \left[ \mathcal{L}_{DDSM}(\overline{\boldsymbol{s}}_{\boldsymbol{\theta}}^{(K+1)}, \lambda(\cdot)) \right] \leq \mathbb{E} \left[ \mathcal{L}_{DDSM}(\overline{\boldsymbol{s}}_{\boldsymbol{\theta}}^{(K)}, \lambda(\cdot)) \right]. \tag{20}$$

*Proof.* Let $K > 1$. As a first step, we establish the following algebraic identity:

$$\frac{1}{K}\sum_{k=1}^{K} \boldsymbol{s}_{\boldsymbol{\theta}}^{(k)}(\mathbf{x}_t, t) = \frac{1}{K}\sum_{j=1}^{K}\frac{1}{K-1}\sum_{k \neq j} \boldsymbol{s}_{\boldsymbol{\theta}}^{(k)}(\mathbf{x}_t, t), \tag{21}$$

which is a direct consequence of

$$\begin{bmatrix} 1 \\ 1 \\ \vdots \\ 1 \\ 1 \end{bmatrix} = \frac{1}{K-1}\left( \begin{bmatrix} 0 \\ 1 \\ \vdots \\ 1 \\ 1 \end{bmatrix} + \begin{bmatrix} 1 \\ 0 \\ \vdots \\ 1 \\ 1 \end{bmatrix} + \cdots + \begin{bmatrix} 1 \\ 1 \\ \vdots \\ 1 \\ 0 \end{bmatrix} \right). \tag{22}$$

We now use Jensen's inequality and the convexity of the norm to obtain the first inequality.

$$\mathcal{L}_{\mathrm{DDSM}}(\bar{\boldsymbol{s}}_{\boldsymbol{\theta}}^{(K)}, \lambda(\cdot)) = \frac{1}{2}\int_0^T \mathbb{E}_{p_0(\mathbf{x}_0)p_{t|0}(\mathbf{x}_t|\mathbf{x}_0)}\left[\lambda(t)\|\left(\frac{1}{K}\sum_{k=1}^{K}\boldsymbol{s}_{\boldsymbol{\theta}}^{(k)}(\mathbf{x}_t, t)\right) - \nabla_{\mathbf{x}_t}\log q_{t|0}(\mathbf{x}_t|\mathbf{x}_0)\|_2^2\right] \tag{23}$$

$$= \frac{1}{2}\int_0^T \mathbb{E}_{p_0(\mathbf{x}_0)p_{t|0}(\mathbf{x}_t|\mathbf{x}_0)}\left[\lambda(t)\|\frac{1}{K}\sum_{j=1}^{K}\frac{1}{K-1}\sum_{k \neq j}\boldsymbol{s}_{\boldsymbol{\theta}}^{(k)}(\mathbf{x}_t, t) - \nabla_{\mathbf{x}_t}\log q_{t|0}(\mathbf{x}_t|\mathbf{x}_0)\|_2^2\right] \tag{24}$$

$$\leq \frac{1}{2K}\sum_{j=1}^{K}\int_0^T \mathbb{E}_{p_0(\mathbf{x}_0)p_{t|0}(\mathbf{x}_t|\mathbf{x}_0)}\left[\lambda(t)\|\frac{1}{K-1}\sum_{k \neq j}\boldsymbol{s}_{\boldsymbol{\theta}}^{(k)}(\mathbf{x}_t, t) - \nabla_{\mathbf{x}_t}\log q_{t|0}(\mathbf{x}_t|\mathbf{x}_0)\|_2^2\right] \tag{25}$$

$$=: \frac{1}{K}\sum_{j=1}^{K}\mathcal{L}_{\mathrm{DDSM}}\left(\frac{1}{K-1}\sum_{k \neq j}\boldsymbol{s}_{\boldsymbol{\theta}}^{(k)},\ \lambda(\cdot)\right). \tag{26}$$

Now assume that for any pair $(\mathbf{x}_t, t)$, the random variables $\boldsymbol{s}^{(1)}(\mathbf{x}_t, t), \ldots, \boldsymbol{s}^{(K)}(\mathbf{x}_t, t)$ are identically distributed given $(\mathbf{x}, t)$. Then,

$$\mathbb{E}[\mathcal{L}_{\mathrm{DDSM}}(\bar{\boldsymbol{s}}_{\boldsymbol{\theta}}^{(K)}, \lambda(\cdot))] = \frac{1}{K}\sum_{j=1}^{K}\mathbb{E}\left[\mathcal{L}_{\mathrm{DDSM}}\left(\frac{1}{K-1}\sum_{k \neq j}\boldsymbol{s}_{\boldsymbol{\theta}}^{(k)},\ \lambda(\cdot)\right)\right] \tag{27}$$

$$= \frac{1}{K}\sum_{j=1}^{K}\mathbb{E}\left[\mathcal{L}_{\mathrm{DDSM}}\left(\frac{1}{K-1}\sum_{k=1}^{K-1}\boldsymbol{s}_{\boldsymbol{\theta}}^{(k)},\ \lambda(\cdot)\right)\right] \tag{28}$$

$$= \mathbb{E}\left[\mathcal{L}_{\mathrm{DDSM}}\left(\bar{\boldsymbol{s}}_{\boldsymbol{\theta}}^{(K-1)}, \lambda(\cdot)\right)\right]. \tag{29}$$

$$\square$$

Similar results are well-established in the case of convex losses (Mattei & Garreau, 2025). Proposition C.1 shows that ensembling multiple score estimators systematically improves the DDSM loss. The first inequality holds deterministically: removing an element from the ensemble uniformly at random will, on average, degrade its performance. The second inequality strengthens this result in expectation, showing that when ordering does not matter (i.e., for i.d. estimators), a larger ensemble performs better than a smaller one in expectation.

We now discuss the i.d. assumption. In our framework, we use DEs, where the models $\boldsymbol{s}_{\boldsymbol{\theta}}^{(1)}, \ldots, \boldsymbol{s}_{\boldsymbol{\theta}}^{(K)}$ are obtained by training the same neural architecture independently according to the same method, on the same

dataset, but starting from different random seeds of the same initialization distribution. As a result, for any fixed input $(\mathbf{x}_t, t)$, the outputs of the individual score models can be regarded as identically distributed samples from an implicit distribution induced by the training randomness. Therefore, our setup satisfies this assumption.

A more general result on score matching can be equivalently deduced. We expect well-trained score models to minimize the following least squares objective:

$$\mathcal{L}_{\text{DSM}}(\boldsymbol{s_\theta}) := \mathbb{E}_{t \sim \mathcal{U}[0,T]} \mathbb{E}_{p_t(\mathbf{x})} \left[ \lambda_t \left\| \nabla_\mathbf{x} \log p_t(\mathbf{x}) - \boldsymbol{s_\theta}(\mathbf{x}, t) \right\|_2^2 \right]. \tag{30}$$

Although this expression is intractable in practice, it is equivalent up to an additive constant to the DDSM loss described in Equation (4), as shown by Vincent (2011). Moreover, by applying the same arguments as in the proof of Proposition C.1, we can establish a similar inequality for Equation (30), namely

$$\mathbb{E}\left[ \mathcal{L}_{\text{DSM}}(\overline{\boldsymbol{s_\theta}}^{(K+1)}) \right] \leq \mathbb{E}\left[ \mathcal{L}_{\text{DSM}}(\overline{\boldsymbol{s_\theta}}^{(K)}) \right]. \tag{31}$$

**$K$ models estimate the path measure better**   Let $q_{0:T}$ and $p_{0:T,\boldsymbol{\theta}}^{\text{SDE}}$ be the path measures of the trajectories $\{\mathbf{x}(t)\}_{t=0}^T$ and $\{\mathbf{x}_{\boldsymbol{\theta}}(t)\}_{t=0}^T$, where the former is a stochastic process solution to Equation (3), and the latter is solution to

$$\mathrm{d}\mathbf{x} = [\mathbf{f}(\mathbf{x}, t) - g(t)^2 \overline{\boldsymbol{s}}_{\boldsymbol{\theta}}^{(K)}(\mathbf{x}, t)]\mathrm{d}t + g(t)\mathrm{d}\bar{\mathbf{w}}. \tag{32}$$

Both measures can be seen as joint distributions for which $q_0$ and $p_{0,\boldsymbol{\theta}}^{\text{SDE}}$ are marginals. Lu et al. (2022) showed that if we consider SDEs with fixed terminal conditions $\mathbf{x}(T) = \mathbf{z}$ and $\mathbf{x}_{\boldsymbol{\theta}}(T) = \mathbf{z}$,

$$D_{KL}(q_{0:T}(\cdot \mid \mathbf{x}(T) = \mathbf{z}) \parallel p_{0:T,\boldsymbol{\theta}}^{\text{SDE}}(\cdot \mid \mathbf{x}_{\boldsymbol{\theta}}(T) = \mathbf{z})) = -\mathbb{E}_{q_{0:T}} \left[ \log \frac{p_{0:T,\boldsymbol{\theta}}^{\text{SDE}}}{q_{0:T}} \right] = \mathcal{L}_{\text{DDSM}}(\overline{\boldsymbol{s}}_{\boldsymbol{\theta}}^{(K)}, g(\cdot)^2). \tag{33}$$

Combining this with Proposition C.1, we show that the aggregation of a large number of score models leads to more precise approximations, on average, of the true SDE solution given a terminal point $\mathbf{z}$.

## C.2   Aggregation rules and their connection to density pooling

In this section, we illustrate how the first three score-level aggregation methods introduced in Section 3.2 can be interpreted through the lens of model composition. The objective is to provide intuition for these aggregation schemes. By model composition, we refer to the combination of multiple probability density functions to form a new distribution, defined up to a normalizing constant. Initially studied for Energy Based Models (EBMs), model composition have mostly been applied for conditional generation in the form of classifier or classifier-free guidance (Du et al., 2023; Skreta et al., 2024; 2025).

**Arithmetic mean of scores.**   Let $p^{(1)}(\mathbf{x}), \ldots, p^{(K)}(\mathbf{x})$ be $K$ probability density functions defined on $\mathbb{R}^d$. Then,

$$\frac{1}{K} \sum_{k=1}^K \nabla_\mathbf{x} \log p^{(k)}(\mathbf{x}) = \nabla_\mathbf{x} \log \sqrt[K]{\prod_{k=1}^K p^{(k)}(\mathbf{x})}. \tag{34}$$

In other words, the arithmetic mean of the individual score functions corresponds to the score of the geometric mean of the densities. This composition yields an unnormalized distribution, but the unknown normalizing constant vanishes under the gradient. Given the central role that the arithmetic mean of score models will play in our study, we formally introduce the geometric mean of densities as a Product of Experts in Definition C.1.

**Definition C.1** (Product of Experts (PoE)). *Let $p^{(1)}(\mathbf{x}), \ldots, p^{(K)}(\mathbf{x})$ be $K$ probability density functions defined on $\mathbb{R}^d$.*

$$\overset{\circ}{p}^{(K)}(\mathbf{x}) = \text{PoE}(p^{(1)}(\mathbf{x}), \ldots, p^{(K)}(\mathbf{x})) := \sqrt[K]{p^{(1)}(\mathbf{z}) \ldots p^{(K)}(\mathbf{z})}/Z_K \tag{35}$$

*where $Z_K = \int \sqrt[K]{p^{(1)}(\mathbf{z}) \ldots p^{(K)}(\mathbf{z})}\mathrm{d}\mathbf{z}$.*

Using AM-GM inequality one can show without difficulty that $Z_K$ is finite and then the normalized composition is well-defined as a density. We now state a result that follows as a direct consequence.

**Sum of scores.** In the same way, we can write the sum of scores as a score since

$$\sum_{k=1}^{K} \nabla_{\mathbf{x}} \log p^{(k)}(\mathbf{x}) = \nabla_{\mathbf{x}} \log \prod_{k=1}^{K} p^{(k)}(\mathbf{x}). \tag{36}$$

The operation $\frac{\prod_{k=1}^{K} p^{(k)}(\mathbf{x})}{\int \prod_{k=1}^{K} p^{(k)}(\mathbf{x}) \mathrm{d}\mathbf{x}}$ was originally called Product of Experts by Hinton (2002). This operation is efficient when the models are different. Indeed it allows each expert to constrain different aspects of the data. As a result, the composed model assigns high probability only to points that satisfy *all* individual constraints simultaneously, like an AND operator. For example, Du et al. (2020) applies the product rule to model the conjunction of concepts with EBMs, where each density is of the form $p^{(k)}(\mathbf{x}) = p(\mathbf{x} \mid c_k)$ with $c_1, \ldots, c_K$ representing different concepts. More recently, Du et al. (2023) applies the product to perform class-conditional and text-to-image generation using diffusion models (e.g. $p^{(k)}(\mathbf{x}) = p(\mathbf{x}|\text{"A sandy beach"})$).

**Mixture of experts.** The equivalent of the Product of Experts as defined above but for the union of distributions is the Mixture of Experts defined by

$$\overline{p}^{(K)}(\mathbf{x}) = \frac{1}{K} \sum_{k=1}^{K} p^{(k)}(\mathbf{x}). \tag{37}$$

This one corresponds to a soft union, assigning high probability to regions where at least one of the experts does. In the case where each $p^{(k)}(\mathbf{x})$ models a conditional distribution $p(\mathbf{x} \mid c_k)$ for a concept $c_k$, the mixture defines a model over samples that belong to *any* of the concepts $\{c_1, \ldots, c_K\}$. A cheap way to model this given $K$ generative models trained for example on different data would be to use a method described in Section 3.2, that is randomly selecting a model (thus a distribution) among the $K$ ones before sampling.

In comparison to the former composition rules, there is no formula that express the score of the mixture only in term of individual scores. We have

$$\nabla_{\mathbf{x}} \log \overline{p}^{(K)}(\mathbf{x}) = \sum_{k=1}^{K} \alpha^{(k)}(\mathbf{x}) \nabla_{\mathbf{x}} \log p^{(k)}(\mathbf{x}) \tag{38}$$

with $\alpha^{(k)}(\mathbf{x}) = \frac{p^{(k)}(\mathbf{x})}{\sum_{j=1}^{K} p^{(j)}(\mathbf{x})}$. It would require to know the individual densities $p^{(1)}(\mathbf{x}), \ldots, p^{(K)}(\mathbf{x})$. Skreta et al. (2024) provides a way to estimate this compositions for diffusion models by using Itô density estimators on the fly during sampling.

Operations on probability densities often correspond to optimal solutions under divergence-based criteria. Typically, the geometric mean is the density $p$ that minimizes the average KL divergence to the individual models

$$\frac{1}{K} \sum_{k=1}^{K} D_{\mathrm{KL}}(p \| p^{(k)}). \tag{39}$$

Similarly, the mixture of experts minimizes the average reverse KL divergence. More broadly, Amari (2007) introduces the concept of "$\alpha$-integration" to generalize density combinations, and establishes a general optimality result for this family of means.

### C.3 Pitfalls of these interpretations

Although the equations described in Appendix C.2 are valid out of the diffusion context, composing score models at each timestep in these ways does not actually guide the diffusion process towards the distribution obtained by reverting the gradient operator and the logarithm. The reasons are the following.

1. **We do not predict a score.** Indeed, generally $s_{\boldsymbol{\theta}}(\mathbf{x}, t) \neq \nabla_{\mathbf{x}} \log p_{t,\boldsymbol{\theta}}^{\mathrm{SDE}}(\mathbf{x})$ and $s_{\boldsymbol{\theta}}(\mathbf{x}, t) \neq \nabla_{\mathbf{x}} \log p_{t,\boldsymbol{\theta}}^{\mathrm{ODE}}(\mathbf{x})$.

2. **Composition and adding noise are not commutative.** Typically, by averaging the scores, we don't sample from a PoE because the distribution at time $t > 0$ is not the PoE of the trained models.

We detail these two observations below.

(1) There is no guarantee that the score model has a structure of a score, or even that is predicts at each step $t$ the score associated to the distribution at time $t$ of the trajectory. For example if we consider the SDE with linear drift, a necessary condition is that $p_{T,\boldsymbol{\theta}}^{\mathrm{SDE}}(\mathbf{x})$ be Gaussian (Lu et al., 2022) which is generally not the case, even though it may be very close to in practice. The reason is that $q_0$ is not Gaussian, and even if it were, the condition would only hold asymptotically as $T \to \infty$. In fact in the ODE case in Equation (13), it has been shown that the score output at time $t = 0$, $\mathbf{s}_{\boldsymbol{\theta}}(\mathbf{x}, t = 0)$, is less accurate than computing the gradient of $p_{0,\boldsymbol{\theta}}^{\mathrm{ODE}}(\mathbf{x})$ with respect to $\mathbf{x}$, as given by Equation (14) (see Feng et al., 2023a, Figure 3). However, well-trained diffusion models can still generate high-quality samples, as we have convergence guarantees: under suitable assumptions, the Wasserstein distance between $p_{0,\boldsymbol{\theta}}^{\mathrm{SDE}}$ and $q_0$ is upper bounded and tends to zero with improved learning and finer Euler discretizations (De Bortoli, 2022).

(2) To sample from the product of densities using diffusion models, for example in the guidance framework (Dhariwal & Nichol, 2021; Ho & Salimans, 2022), we operate the product at each noise level of the sampling procedure by replacing the score with what would be the score of the product. For instance, given a feature $\mathbf{y}$ (e.g. a class), to estimate $p(\mathbf{x}|\mathbf{y})$, we sample from a slighly different distribution

$$\underbrace{\widetilde{p}(\mathbf{x} \mid \mathbf{y})}_{\overset{\circ}{p}{}^{(2)}(\mathbf{x})} \propto \underbrace{p(\mathbf{x})}_{p^{(1)}(\mathbf{x})} \underbrace{p(\mathbf{y} \mid \mathbf{x})^{1+w}}_{p^{(2)}(\mathbf{x})} \tag{40}$$

$$= p(\mathbf{x}) \left( \frac{p(\mathbf{x} \mid \mathbf{y}) \, p(\mathbf{y})}{p(\mathbf{x})} \right)^{1+w} \tag{41}$$

$$\propto p(\mathbf{x}) \left( \frac{p(\mathbf{x} \mid \mathbf{y})}{p(\mathbf{x})} \right)^{1+w} \tag{42}$$

$$= \underbrace{p(\mathbf{x})^{-w}}_{p^{(1)}(\mathbf{x})} \underbrace{p(\mathbf{x} \mid \mathbf{y})^{1+w}}_{p^{(2)}(\mathbf{x})} \tag{43}$$

where $w$ is a temperature parameter. While Dhariwal & Nichol (2021) rely on Equation (40) and train a classifier independently from the diffusion model to approximate the conditional likelihood, Ho & Salimans (2022) adopt Equation (43) and jointly train two diffusion models instead. Hence, if for example we consider the latter, we compute at test time $\nabla_{\mathbf{x}} \log \tilde{p}_t(\mathbf{x} \mid \mathbf{y}) = -w \nabla_{\mathbf{x}} \log p_t(\mathbf{x}) + (1 + w) \nabla_{\mathbf{x}} \log p_t(\mathbf{x}|\mathbf{y})$ for every $t > 0$, where $p_t(\mathbf{x}|\mathbf{y})$ and $\nabla_{\mathbf{x}} \log p_t(\mathbf{x})$ are independently learned. However, this stepwise rule assume that the proportionality in Equation (43) hold for noise levels $t > 0$. It is false when $p_t$ and $p_t(\cdot \mid \mathbf{y})$ are the diffusion distributions obtained by computing convolution of $p$ and $p(\cdot \mid \mathbf{y})$ with Gaussian noise levels. Fortunately, this pitfall does not compromise the effectiveness of the guidance procedure in practice, showing that this trick is a good proxy of the target distribution.

As indicated by Du et al. (2023) and Chidambaram et al. (2024), the misconception in reasoning is also true in the general case of product and geometric mean: the operations of applying noise to the probabilities and computing the product or the PoE in Definition C.1 are not commutative. We may schematically write

$$\mathrm{PoE}(p^{(1)}(\mathbf{x}), \dots, p^{(K)}(\mathbf{x}))_t \neq \mathrm{PoE}(p_t^{(1)}(\mathbf{x}), \dots, p_t^{(K)}(\mathbf{x})) \tag{44}$$

as soon as $t > 0$ if $p^{(1)}(\mathbf{x}), \dots, p^{(K)}(\mathbf{x})$ are the starting distributions associated to each trained model (in fact $\neq$ means "not proportional to" here). The direct consequence is

$$\nabla_{\mathbf{x}} \log \overset{\circ}{p}{}_t^{(K)}(\mathbf{x}) \neq \frac{1}{K} \sum_{k=1}^{K} \log p_t^{(k)}(\mathbf{x}). \tag{45}$$

However, to the best of our knowledge, no simple, closed-form counterexample has been found to support this inequality. In the following section, we provide a proof using the case of centered Gaussians.

### C.4 Diffused PoE is not the PoE of diffused distributions: proof in the Gaussian case

In this part we show that in general, the PoE of distributions at $t = 0$ does not lead to the PoE of marginal distributions $\{p_t^{(k)}\}_{t>0}^T$ induced by each of the diffusion trajectories. More precisely, we demonstrate that a necessary and sufficient condition is that are all initial distributions are equal (e.g. we sample a unique distribution). We prove this assuming the target distributions are all centered Gaussians in $\mathbb{R}^d$. This case is particularly interesting, as it yields an explicit score and admits analytical solutions for the backward SDE (Pierret & Galerne, 2025).

Following Pierret & Galerne (2025), we consider the Variance Preserving forward SDE (or Ornstein–Uhlenbeck process)

$$d\mathbf{z}_t = -\beta_t \mathbf{z}_t dt + \sqrt{2\beta_t} d\mathbf{w}_t, \quad 0 \leq t \leq T, \quad \mathbf{z}_0 \sim q_0. \tag{46}$$

The distribution $q_0$ is noised progressively according to the variance schedule $\beta_t$. The equation 46 admits one strong solution written as

$$\mathbf{z}_t = \gamma_t \mathbf{z}_0 + \eta_t, \quad 0 \leq t \leq T \tag{47}$$

where $\gamma_t \in (0,1)$ and $\eta_t > 0$ are respectively deterministic and gaussian independent of $\mathbf{z}_0$. However both depend on $\beta_t$. Moreover, if $\Sigma_t$ denotes the covariance matrix of $\mathbf{z}_t$, we have

$$\mathbf{\Sigma}_t = \gamma_t \mathbf{\Sigma} + (1 - \gamma_t)\boldsymbol{I} \tag{48}$$

if $\mathbf{\Sigma}$ is the covariance matrix of $\mathbf{z}_0$.

**Assumption 1** (Gaussian assumption). In the following we assume that $q_0$ is a centered Gaussian distribution, namely $\mathcal{N}(\mathbf{0}, \mathbf{\Sigma})$, and that $\mathbf{\Sigma}$ is invertible. In this case,

$$\mathbf{z}_t \sim q_t = \mathcal{N}(\mathbf{0}, \mathbf{\Sigma}_t) \tag{49}$$

and $\mathbf{\Sigma}_t$ is invertible for $t > 0$.

**What is the PoE of (centered) Gaussians?** Assume $p^{(1)}, \dots, p^{(K)}$ are all Gaussian distributions equal to $\mathcal{N}(\mathbf{0}, \mathbf{\Sigma}_k)$. Then

$$\mathrm{PoE}(p^{(1)}, \dots, p^{(K)}) = \mathcal{N}\left(\mathbf{0}, \left(\frac{1}{K}\sum_{k=1}^K \mathbf{\Sigma}_k^{-1}\right)^{-1}\right) \tag{50}$$

where the equality holds thanks to the normalization coefficient.

**Main proposition.** We start with the following lemma.

**Lemma C.1.** *Let $\mathbf{a} \in \mathbb{R}_{>0}^K$ and $\mathbf{b} \in \mathbb{R}_{>0}^K$. Let $H : (x_1, \dots, x_K) \mapsto (\frac{1}{K}\sum_{k=1}^K x_k^{-1})^{-1}$ denote the harmonic mean function that takes a vector in $\mathbb{R}_{>0}^K$. Then $H$ is super-additive:*

$$H(\mathbf{a}) + H(\mathbf{b}) \leq H(\mathbf{a} + \mathbf{b}) \tag{51}$$

*and equality holds if and only if there exists $\lambda > 0$ such that $\mathbf{a} = \lambda \mathbf{b}$ (we write $\mathbf{a} \propto \mathbf{b}$).*

*Proof.* Let $\mathbf{a} = (a_1, \dots, a_K) \in \mathbb{R}_{>0}^K$ and $\mathbf{b} = (b_1, \dots, b_K) \in \mathbb{R}_{>0}^K$. The inequality corresponds to a special case of the reverse Minkowski's Inequality for Sums (Proposition C.3)

$$\left(\sum_{k=1}^K (a_k + b_k)^p\right)^{1/p} \geq \left(\sum_{k=1}^K a_k^p\right)^{1/p} + \left(\sum_{k=1}^K b_k^p\right)^{1/p}, \tag{52}$$

valid for all $p < 1$, and in particular for $p = -1$. $\qquad\square$

The counter-example is the following.

**Proposition C.2.** *Assume we are under Gaussian assumption for $K > 1$ distributions $p_0^{(1)}, \ldots, p_0^{(K)}$, and moreover each initial covariance matrix is equal to $\alpha_k \boldsymbol{I}$ with $\alpha_k > 0$. If $p_0 = PoE(p_0^{(1)}, \ldots, p_0^{(K)})$ is the initial condition of Eq. (46), then for each $t > 0$, the marginal distributions $p_t$ associated to the strong solution $\{\mathbf{x}_t\}_{t>0}^T$ verifies*

$$p_t = PoE(p_t^{(1)}, \ldots, p_t^{(K)}) \tag{53}$$

*if and only if $\alpha_1 = \cdots = \alpha_K$.*

*Proof.* Let $K > 1$, $0 < t \leq T$, and $\alpha_1, \ldots, \alpha_K > 0$. Let us derive the distribution $p_t$. From Eq. (49) and Eq. (50), $p_t = \mathcal{N}(\mathbf{0}, \boldsymbol{\Sigma}_t^{\mathrm{PoE}})$ where

$$\boldsymbol{\Sigma}_t^{\mathrm{PoE}} = \gamma_t \Big(\frac{1}{K} \sum_{k=1}^{K} \alpha_k^{-1} \boldsymbol{I}\Big)^{-1} + (1 - \gamma_t)\boldsymbol{I} = \underbrace{\left(\gamma_t \Big(\frac{1}{K} \sum_{k=1}^{K} \alpha_k^{-1}\Big)^{-1} + (1 - \gamma_t)\right)}_{c_t^{\mathrm{PoE}}} \boldsymbol{I}. \tag{54}$$

In the other side, $\mathrm{PoE}(p_t^{(1)}, \ldots, p_t^{(K)}) = \mathcal{N}(\mathbf{0}, \boldsymbol{\Sigma}_t^{\mathrm{not\ PoE}})$ where

$$\boldsymbol{\Sigma}_t^{\mathrm{not\ PoE}} = \left(\frac{1}{K} \sum_{k=1}^{K} (\gamma_t \alpha_k \boldsymbol{I} + (1 - \gamma_t)\boldsymbol{I})^{-1}\right)^{-1} = \underbrace{\left(\frac{1}{K} \sum_{k=1}^{K} (\gamma_t \alpha_k + (1 - \gamma_t))^{-1}\right)^{-1}}_{c_t^{\mathrm{not\ PoE}}} \boldsymbol{I}. \tag{55}$$

Let us compare $\boldsymbol{\Sigma}_t^{\mathrm{PoE}}$ and $\boldsymbol{\Sigma}_t^{\mathrm{not\ PoE}}$. We establish the following result:

$$c_t^{\mathrm{PoE}} \leq c_t^{\mathrm{not\ PoE}} \tag{56}$$

and $c_t^{\mathrm{PoE}} = c_t^{\mathrm{not\ PoE}} \Leftrightarrow \forall k \in \{1, \ldots, K\}, \quad \alpha_k = \lambda_t \frac{(1 - \gamma_t)}{\gamma_t}$ with $\lambda_t > 0$.

To prove this, we write both scalar values in terms of harmonic means.

$$\begin{cases} c_t^{\mathrm{PoE}} &= \gamma_t H(\alpha_1, \ldots, \alpha_K) + (1 - \gamma_t) \\ c_t^{\mathrm{not\ PoE}} &= H(\gamma_t \alpha_1 + (1 - \gamma_t), \ldots, \gamma_t \alpha_K + (1 - \gamma_t)) \end{cases} \tag{57}$$

where $H : \mathbf{x} = (x_1, \ldots, x_K) \mapsto (\frac{1}{K} \sum_{k=1}^{K} x_k^{-1})^{-1}$. Since everything is positive, the result is straightforward using Lemma C.1 and $H(\lambda \mathbf{x}) = \lambda H(\mathbf{x})$ for all $(\lambda, \mathbf{x}) \in (\mathbb{R}_{>0}, \mathbb{R}_{>0}^K)$.

$$c_t^{\mathrm{PoE}} = \gamma_t H(\alpha_1, \ldots, \alpha_K) + (1 - \gamma_t)H(1, \ldots, 1) \tag{58}$$
$$= H(\gamma_t(\alpha_1, \ldots, \alpha_K)) + H((1 - \gamma_t)(1, \ldots, 1)) \tag{59}$$
$$\leq H(\gamma_t \alpha_1 + (1 - \gamma_t), \ldots, \gamma_t \alpha_K + (1 - \gamma_t)) \tag{60}$$
$$= c_t^{\mathrm{not\ PoE}} \tag{61}$$

with equality if and only there exists $\lambda_t > 0$ such that for all $k \in \{1, \ldots, K\}$, $\alpha_k = \lambda_t \frac{(1 - \gamma_t)}{\gamma_t}$, which is equivalent to all the $\alpha_k$ being equal. $\qquad\square$

This proof demonstrates that in the particular Gaussian case, the equality is equivalent to all the individual starting distributions being identical. In this case, and even more generally, for any $t \in (0, T]$ the intermediate probability associated to $t$ cannot be expected to follow a PoE structure, and computing the PoE at intermediate steps does not faithfully reflect the PoE of the initial distributions.

## C.5 Implications of non-commutativity in two contexts

We provide two well-known yet simple frameworks that are affected by Proposition 4.2: concept-conditional generative modeling and linear inverse problems.

**Concept-conditional generative modeling.** In this setting (see Appendix C.3), the components of the product are expected to be different models. Thus, Proposition C.2 hints that relying solely on the guidance approach is unlikely to produce samples from the true target conditional distribution. Some recent works leverage some hacks like MCMC or Feynman-Kac-based corrections during sampling to handle this theoretical pitfall and improve sampling accuracy (Du et al., 2023; Skreta et al., 2025). No such correctors are applied in our experiments, as we focus on the baseline ensembling behavior without additional guidance refinements.

**Linear inverse problems.** Let us consider linear inverse problems, that is, we observe a measurement

$$\mathbf{y} = \boldsymbol{A}\mathbf{x}_0 + \boldsymbol{\epsilon}, \tag{62}$$

where $\boldsymbol{A} \in \mathbb{R}^{m \times d}$ is a known linear operator (or measurement matrix), $\mathbf{x}_0 \in \mathbb{R}^d$ is the unknown signal to be recovered, and $\boldsymbol{\epsilon} \in \mathbb{R}^m$ denotes additive noise, typically assumed to be a centered Gaussian variable independent of $\mathbf{x}_0$. The goal is to recover $\mathbf{x}_0$ from the observation $\mathbf{y}$, possibly under prior assumptions on the distribution of $\mathbf{x}_0$ or through a generative model. We would like to estimate $p(\mathbf{x}|\mathbf{y}) \propto p(\mathbf{x})p(\mathbf{y}|\mathbf{x})$. We can again adopt the "guidance" approach by using a pre-trained model aiming to estimate $\nabla_{\mathbf{x}} \log p_t(\mathbf{x}_t)$ and add it to an approximation of the measurement matching term $\nabla_{\mathbf{x}} \log p_t(\mathbf{y}|\mathbf{x}_t)$ for each noise level of the sampling process. The latter term is estimated using the identity

$$p_t(\mathbf{y} \mid \mathbf{x}_t) = \int p(\mathbf{y} \mid \mathbf{x}_0) \, p(\mathbf{x}_0 \mid \mathbf{x}_t) \, \mathrm{d}\mathbf{x}_0. \tag{63}$$

and numerous approximation methods can be leveraged (Chung et al., 2023; Song et al., 2023).

Assume $\mathbf{x} \coloneqq \mathbf{x}_0 \sim q_0 = \mathcal{N}(\mathbf{0}, \boldsymbol{\Sigma})$. For this part, let us consider *e.g.* $m = d$ and $\boldsymbol{A} \succ 0$. The likelihood function $p(\mathbf{y}|\mathbf{x})$ taken in $\mathbf{x}$ has a quadratic logarithm and corresponds to

$$p(\mathbf{y}|\mathbf{x}) \propto \exp\Big(-\frac{1}{2}(\mathbf{y} - \boldsymbol{A}\mathbf{x})^\top \boldsymbol{\Sigma}^{-1}(\mathbf{y} - \boldsymbol{A}\mathbf{x})\Big) \tag{64}$$

$$\propto \exp\Big(-\frac{1}{2}\big(\mathbf{x}^\top \underbrace{(\boldsymbol{A}^\top \boldsymbol{\Sigma}^{-1} \boldsymbol{A})}_{\widehat{\boldsymbol{\Sigma}}^{-1}} \mathbf{x} - 2\mathbf{x}^\top (\boldsymbol{A}^\top \boldsymbol{\Sigma}^{-1} \mathbf{y})\big)\Big) \tag{65}$$

$$\propto \exp\Big(-\frac{1}{2}\big(\mathbf{x}^\top \widehat{\boldsymbol{\Sigma}}^{-1}\mathbf{x} - 2\mathbf{x}^\top \widehat{\boldsymbol{\Sigma}}^{-1} \underbrace{\widehat{\boldsymbol{\Sigma}} (\boldsymbol{A}^\top \boldsymbol{\Sigma}^{-1} \mathbf{y})}_{\widehat{\mu}}\big)\Big) \tag{66}$$

$$\propto \exp\Big(-\frac{1}{2}(\mathbf{x} - \widehat{\mu})^\top \widehat{\boldsymbol{\Sigma}}^{-1}(\mathbf{x} - \widehat{\mu})\Big). \tag{67}$$

This yields a Gaussian form with mean $(\boldsymbol{A}^\top \boldsymbol{\Sigma}^{-1} \boldsymbol{A})^{-1}\boldsymbol{A}^\top \boldsymbol{\Sigma}^{-1}\mathbf{y}$ and covariance $(\boldsymbol{A}^\top \boldsymbol{\Sigma}^{-1} \boldsymbol{A})^{-1}$, provided the latter is positive definite (which is the case here since $m = d$ and $\boldsymbol{A} \succ 0$). When $\mathbf{y} = \mathbf{0}$, the distribution is centered. Because $p(\mathbf{x})$ and $p(\mathbf{y} \mid \mathbf{x})$ are distinct, in particular they have different covariances, the necessary condition in Proposition C.2 is violated, and therefore we cannot expect Equation (53) to hold in this setting.

## C.6 Some useful algebra results

**Lemma C.2** (Reverse Young's Inequality for Products). *Let $p, q \in \mathbb{R}_{>0}$ be strictly positive real numbers satisfying: $\frac{1}{p} - \frac{1}{q} = 1$ Let $a \in \mathbb{R}_{\geq 0}$ be a positive real number and $b \in \mathbb{R}_{>0}$ be a strictly positive real number. Then*

$$ab \geq \frac{a^p}{p} - \frac{b^q}{q}. \tag{68}$$

*with equality if and only if $a^p = b^{-q}$.*

*Proof.* We define $u$ and $v$ such that $\frac{1}{u} = p$ and $\frac{1}{v} = \frac{p}{q}$. By hypothesis, $\frac{1}{u} + \frac{1}{v} = 1$, thus we can apply Young's Inequality for Products (Rudin, 1976).

$$(ab)^p b^{-p} \leq \frac{((ab)^p)^{1/p}}{1/p} + \frac{(b^{-p})^{q/p}}{q/p} \tag{69}$$

$$\implies \quad a^p \leq pab + p\frac{b^{-q}}{q} \tag{70}$$

$$\implies \quad \frac{a^p}{p} \leq ab + \frac{b^{-q}}{q} \tag{71}$$

$$\implies \quad ab \geq \frac{a^p}{p} - \frac{b^{-q}}{q}. \tag{72}$$

Equality holds if and only if the equality case in Young's inequality is attained, that is when

$$ab = b^{-q} \iff a^p = b^{-q}. \tag{73}$$

$\square$

**Lemma C.3** (Reverse Hölder's Inequality for Sums). *Let $p, q \in \mathbb{R}_{>0}$ be strictly positive real numbers such that $\frac{1}{p} - \frac{1}{q} = 1$.*

*Suppose that the sequences $\mathbf{x} = \{x_n\}_{n \in \mathbb{N}}$ and $\mathbf{y} = \{y_n\}_{n \in \mathbb{N}}$ in $\mathbb{R}$ or $\mathbb{R}^d$ are such that the series*

$$\|\mathbf{x}\|_p := \left( \sum_{n=1}^{\infty} |x_n|^p \right)^{1/p} \tag{74}$$

*and*

$$\|\mathbf{y}\|_{-q} := \left( \sum_{n=1}^{\infty} |y_n|^{-q} \right)^{-1/q} \tag{75}$$

*are convergent. Let $\|\mathbf{xy}\|_1$ denote the 1-norm of $\mathbf{xy}$, if $\mathbf{xy}$ is in the Lebesgue space $\ell^1$. Then,*

$$\|\mathbf{xy}\|_1 \geq \|\mathbf{x}\|_p \|\mathbf{y}\|_{-q}. \tag{76}$$

*Equality holds if and only if there exists a constant $c > 0$ such that for all $n \in \mathbb{N}$,*

$$|x_n|^p = c \cdot |y_n|^{-q}. \tag{77}$$

*Proof.* Without loss of generality, assume that $\mathbf{x}$ and $\mathbf{y}$ are non-zero.

Let

$$\mathbf{u} = \{u_n\}_{n \in \mathbb{N}} = \frac{\mathbf{x}}{\|\mathbf{x}\|_p} \tag{78}$$

and

$$\mathbf{v} = \{v_n\}_{n \in \mathbb{N}} = \frac{\mathbf{y}}{\|\mathbf{y}\|_{-q}}. \tag{79}$$

Then,

$$\|\mathbf{u}\|_p = \|\mathbf{v}\|_{-q} = 1. \tag{80}$$

Since $\frac{1}{p} - \frac{1}{q} = 1$, by Reverse Young's Inequality for Products we have (Lemma C.2),

$$|u_n v_n| \geq \frac{1}{p}|u_n|^p - \frac{1}{q}|v_n|^{-q}. \tag{81}$$

and summing over all $n \in \mathbb{N}$ gives

$$\|\mathbf{uv}\|_1 \geq \frac{1}{p}\|\mathbf{u}\|_p^p - \frac{1}{q}\|\mathbf{v}\|_{-q}^{-q} = 1 \tag{82}$$

as desired.

From Lemma C.2, equality holds if and only if

$$\forall n \in \mathbb{N}, |u_n|^p = |v_n|^{-q}, \tag{83}$$

and then Hölder's inequality becomes an inequality if and only if there exists $\alpha, \beta > 0$ (namely $\alpha = \|\mathbf{y}\|_{-q}^{-q}$ and $\beta = \|\mathbf{x}\|_p^p$ ) such that

$$\alpha|x_n|^p = \beta|y_n|^{-q}. \tag{84}$$

$\square$

**Proposition C.3** (Minkowski's Reverse Inequality for Sums: case $p < 1$). *Let* $\mathbf{a} = (a_1, \ldots, a_K)^T \in \mathbb{R}_{>0}^K$, $\mathbf{b} = (b_1, \ldots, b_K)^T \in \mathbb{R}_{>0}^K$, *and* $p \in \mathbb{R}$.

*If* $p < 1$, $p \neq 0$, *then*

$$\left(\sum_{k=1}^K (a_k + b_k)^p\right)^{1/p} \geq \left(\sum_{k=1}^K a_k^p\right)^{1/p} + \left(\sum_{k=1}^K b_k^p\right)^{1/p} \tag{85}$$

*and equality holds if and only if* $\mathbf{a} = c \cdot \mathbf{b}$ *for some* $c > 0$.

*Proof.* Define $q = \frac{p}{p-1}$. Then,

$$\frac{1}{p} - \frac{1}{-q} = \frac{1}{p} + \frac{p-1}{p} = 1 \tag{86}$$

where $p > 0$ and $-q > 0$. Using Lemma C.3 followed by $(p-1)q = p$, we have

$$\sum_{k=1}^K (a_k + b_k)^p = \sum_{k=1}^K a_k(a_k + b_k)^{p-1} + \sum_{k=1}^K b_k(a_k + b_k)^{p-1} \tag{87}$$

$$\geq \left(\sum_{k=1}^K a_k^p\right)^{1/p} \left(\sum_{k=1}^K \left((a_k + b_k)^{p-1}\right)^q\right)^{1/q} + \left(\sum_{k=1}^K b_k^p\right)^{1/p} \left(\sum_{k=1}^K \left((a_k + b_k)^{p-1}\right)^q\right)^{1/q} \tag{88}$$

$$= \left(\sum_{k=1}^K a_k^p\right)^{1/p} \left(\sum_{k=1}^K (a_k + b_k)^p\right)^{1/q} + \left(\sum_{k=1}^K b_k^p\right)^{1/p} \left(\sum_{k=1}^K (a_k + b_k)^p\right)^{1/q} \tag{89}$$

$$= \left[\left(\sum_{k=1}^K a_k^p\right)^{1/p} + \left(\sum_{k=1}^K b_k^p\right)^{1/p}\right] \left(\sum_{k=1}^K (a_k + b_k)^p\right)^{1/q}. \tag{90}$$

$$\tag{91}$$

Then,

$$\left(\sum_{k=1}^K (a_k + b_k)^p\right)^{1-1/q} \geq \left(\sum_{k=1}^K a_k^p\right)^{1/p} + \left(\sum_{k=1}^K b_k^p\right)^{1/p} \tag{92}$$

$$\Rightarrow \left(\sum_{k=1}^K (a_k + b_k)^p\right)^{1/p} \geq \left(\sum_{k=1}^K a_k^p\right)^{1/p} + \left(\sum_{k=1}^K b_k^p\right)^{1/p}. \tag{93}$$

From Lemma C.3, equality case holds if and only if there exists $c_1, c_2 > 0$ such that $a_k^p = c_1 \cdot (a_k + b_k)^p$ and $b_k^p = c_2(a_k + b_k)^p$, which means $\mathbf{a} \propto \mathbf{b}$. To demonstrate the latter, suppose $a_k^p = c_1(a_k + b_k)^p$ and $b_k^p = c_2(a_k + b_k)^p$ for all $k$, with $c_1, c_2 > 0$. Then

$$\left(\frac{a_k}{a_k + b_k}\right)^p = c_1, \quad \left(\frac{b_k}{a_k + b_k}\right)^p = c_2. \tag{94}$$

Dividing the two equations yields

$$\left(\frac{a_k}{b_k}\right)^p = \frac{c_1}{c_2} \quad \Rightarrow \quad \frac{a_k}{b_k} = \left(\frac{c_1}{c_2}\right)^{1/p} \quad \text{for all } k.$$

Hence, $\mathbf{a} \propto \mathbf{b}$. Conversely, if $\mathbf{a} = c \cdot \mathbf{b}$ for some $c > 0$, then $a_k + b_k = (1 + c) \cdot b_k$, so

$$a_k^p = c^p \cdot b_k^p = c^p (1 + c)^{-p} \cdot (a_k + b_k)^p,$$
$$b_k^p = (1 + c)^{-p} \cdot (a_k + b_k)^p.$$

Thus, the condition holds with $c_1 = c^p (1 + c)^{-p}$ and $c_2 = (1 + c)^{-p}$.

$\square$

## D   Additional aggregation schemes

While Section 3.2 mainly focuses on step-wise aggregation schemes, it includes only one model that operates at a higher level. This raises the question of whether ensembling can be performed solely at the final stage of the process, for example. Additionally, all our step-wise aggregators follow the form SDE_SOLVER(AGG($\cdot, \ldots, \cdot$)), as illustrated in Figure 2, but one may wonder what happens if we reverse the order and apply the AGG function after each step of the SDE SOLVER. We describe two additional aggregation schemes in Appendix D.1 and show in Appendix D.2 that these methods do not perform better in practice on Deep Ensemble than our main approaches.

### D.1   Detailed descriptions

We list two methods that operate differently on the sampling trajectories. Song et al. (2021b) provides a general framework for reverse diffusion sampling by combining for each noise level a predictor step, that drives the sample toward the solution of the SDE, and a corrector step that drives the sample to a more correct sample given the noise level. We write $\text{Predictor}(\mathbf{x}_{t_n}, t_n) = \gamma_n \mathbf{x}_{t_n} + \tilde{P}(\boldsymbol{s_\theta}(\mathbf{x}_{t_n}, t_n), t_n, \mathbf{z}_n)$ and $\text{Corrector}(\mathbf{x}_{t_n}) = \delta_n \mathbf{x}_{t_n} + \tilde{C}(\boldsymbol{s_\theta}(\mathbf{x}_{t_n}, t_n), \mathbf{z}_n)$ where $\mathbf{z}_n$ is a standard gaussian noise. If we consider Euler-Maruyama step as predictor, $\gamma_n = 1$ and $\tilde{P}(\boldsymbol{s_\theta}(\mathbf{x}_{t_n}, t_n), t_n, \mathbf{z}_n) = \left[\mathbf{f}(\mathbf{x}_{t_n}, t_n) - g^2(t_n) \boldsymbol{s_\theta}(\mathbf{x}_{t_n}, t_n)\right] \Delta t + g(t_n)\sqrt{\Delta t}\mathbf{z}_n$. For DDPM, we construct a sequence of scales $(\beta_{t_n})_{n \in \{1, \ldots, N\}}$, then set $\gamma_n = \frac{1}{\sqrt{1 - \beta_{t_n}}}$, and $\tilde{P}(\boldsymbol{s_\theta}(\mathbf{x}_{t_n}, t_n), t_n, \mathbf{z}_n) = \frac{\beta_{t_n}}{\sqrt{1 - \beta_{t_n}}} \boldsymbol{s_\theta}(\mathbf{x}_{t_n}, t_n) + \sqrt{\beta_{t_n}}\mathbf{z}_n$. If we consider Langevin dynamics for the corrector, we can set $\delta_n = 1$ and $\tilde{C}(\boldsymbol{s_\theta}(\mathbf{x}_{t_n}, t_n), \mathbf{z}_n) = \epsilon_n \boldsymbol{s_\theta}(\mathbf{x}_{t_n}, t_n) + \sqrt{2\epsilon_n}\mathbf{z}_n$.

**Average of noises.** At each step $t_{n+1}$, we can average the noises added to $\gamma_n \mathbf{x}_{t_n}$ to get $\mathbf{x}_{t_{n+1}}$ in the predictor. Let's define $\Delta_{n+1} \coloneqq \mathbf{x}_{t_{n+1}} - \delta_n \mathbf{x}_{t_n}$. It represents the noise injected into the image state to advance to the next step. Here we propose $\frac{1}{K}\sum_{k=1}^{K} \Delta_{n+1}^{(k)} = \frac{1}{K}\sum_{k=1}^{K} \tilde{P}(\boldsymbol{s_\theta}^{(k)}(\mathbf{x}_{t_n}, t_n), t_n, \mathbf{z}_n^{(k)})$, assuming for simplicity that only the Predictor step is taken into account. Typically, if we consider Euler-Maruyama steps, the update corresponds to

$$\left[\mathbf{f}(\mathbf{x}_{t_n}, t_n) - g^2(t_n)\overline{\boldsymbol{s}}_{\boldsymbol{\theta}}^{(K)}(\mathbf{x}_{t_n}, t_n)\right] \Delta t + g(t_n)\sqrt{\Delta t}\frac{1}{K}\sum_{k=1}^{K} \mathbf{z}_n^{(k)} \tag{95}$$

which is almost equivalent to simply averaging the scores due to the linearity of the updates, though it differs because the Gaussian noise terms are also averaged. Not that the latter vanish asymptotically with $K$, leaving only the contribution of the averaged scores. Asymptotically it corresponds to the discretization of an ODE

$$d\mathbf{x} = [\mathbf{f}(\mathbf{x}, t) - g(t)^2 \mathbb{E}_{k \sim \mathcal{U}[\![1,K]\!]}[\boldsymbol{s}_{\boldsymbol{\theta}}^{(k)}(\mathbf{x}, t)]dt + g(t)\underbrace{\mathbb{E}[d\bar{\mathbf{w}}]}_{=0}. \tag{96}$$

One can also average the corrections in the corrector steps, similarly to the predictor.

**Mean of predictions / Posterior mean approximation.** We compute the mean over all samples $\widehat{\mathbf{x}}_0$ generated from $K$ sampling paths for a given input $\mathbf{z}$. This approach aligns with the objective of Bayesian inference in the generative modeling framework. For diffusion models, the posterior predictive distribution is obtained from an initial noise $\mathbf{z}$ following a "prior" distribution (Gaussian noise). Hence averaging the outputs corresponds to approximating

$$p(\widehat{\mathbf{x}}_0|\mathbf{z}, \mathcal{D}) = \int p(\widehat{\mathbf{x}}_0|\mathbf{z}, \boldsymbol{\theta})q(\boldsymbol{\theta}|\mathcal{D})d\boldsymbol{\theta} \approx \frac{1}{K}\sum_{k=1}^{K} p(\widehat{\mathbf{x}}_0|\mathbf{z}, \boldsymbol{\theta}_k) \tag{97}$$

The difference from our framework is that $\mathbf{z}$ represents here the stochasticity introduced in the entire sampling process (not only the starting noise). Some works employ this bayesian inference technique, using different ensemble strategies to sample $\boldsymbol{\theta}_k$ like Laplace approximation (Ritter et al., 2018) or Hyper Networks (Ha et al., 2017), to estimate uncertainty for diffusion models Chan et al. (2024); Jazbec et al. (2025).

## D.2 These methods are under-performing

We present additional results complementing Section 5.2.1, by incorporating aggregation schemes that perform extremely poorly and are described in Appendix D.1. In addition to the sum mentioned in Section 3.2, we also evaluate the two trajectory-level aggregation schemes described in Appendix D. We provide results on Deep Ensemble using ADM models trained on FFHQ.

Table 4: Comparison of different aggregation methods for on FID, IS, and KID. The parameter $\eta$ defines the level of entropy involved during DDIM generation, see Song et al. (2020) for a clear definition. The result highlighted in orange is the one reported in Table 1.

| | CIFAR-10 ($32 \times 32$) | | | FFHQ ($256 \times 256$) | |
|---|---|---|---|---|---|
| Deep Ensemble | FID $\downarrow$ | IS $\uparrow$ | KID $\downarrow$ | FID $\downarrow$ | KID $\downarrow$ |
| Individual model DE | **4.79** (0.08) | **9.37** (0.07) | **0.001** (0.000) | **24.20** (5.70) | **0.015** (0.003) |
| Sum of scores | 516.19 | 1.14 | 0.591 | 524.33 | 0.69 |
| Mean predictions ($\eta$=0.0) | — | — | — | 58.31 | 0.04 |
| Mean predictions ($\eta$=0.5) | — | — | — | 144.89 | 0.13 |
| Best model | 4.65 | 9.45 | 0.0006 | 21.7 | 0.012 |

## E Tables and plots for U-Nets ensembles

In this section, we first present in Figure 8 the distribution of individual FID-10k values to complement Table 1. We also provide in Tables 5 and 6 and fig. 10 additional tables and plots on ensemble strategies to complement our study in neural network ensembling for diffusion models (Sections 5.2.1 to 5.2.3). We also report the effect on NLL given by increasing $K$ in Figure 9.

### E.1 Distribution of FID-10k from DE on CIFAR-10

### E.2 Aggregation schemes

### E.3 Ensembling on the noise space

We further investigate in Table 7 whether limiting model averaging to the early stages of the reverse diffusion process can mitigate the degradation in perceptual quality observed with full-step ensembling. The rationale is that, during the initial timesteps, the model primarily reconstructs coarse structures, while later steps refine high-frequency details that strongly influence perceptual metrics such as FID. Averaging may thus be less harmful if applied only to the early denoising stages. To test this hypothesis, we perform ensembling over the first third of the diffusion steps only, keeping the remaining steps deterministic.

Partial ensembling does not improve FID compared to individual models. Interestingly, the resulting performance (FID = 4.83) lies between that of the individual models and full-step ensembling, suggesting that the

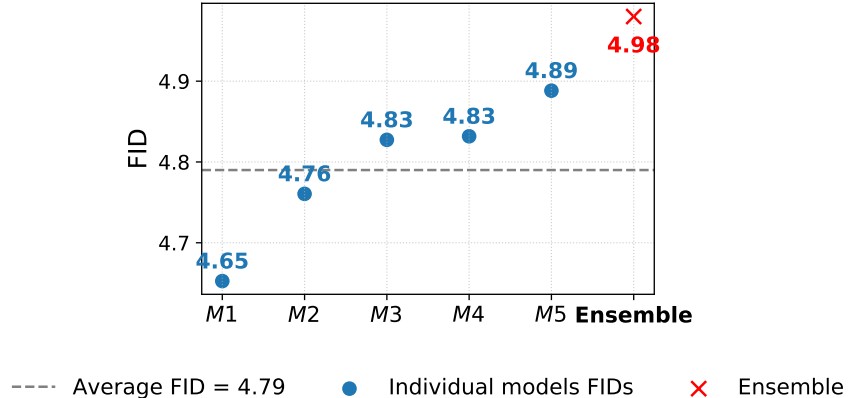

Figure 8: FID-10k values from all individual models M$i$ trained with DE where $i \in \{1, 2, 3, 4, 5\}$. Ensemble aggregation is the arithmetic mean. Score averaging performs worse than the weakest individual model, which suggests an existing blurring effect caused by score averaging at all timesteps. It illustrates that ensembling can sometimes be detrimental.

Table 5: Comparison of different aggregation methods for $K = 20$ on CIFAR-10 in terms of FID, KID, and $L_{\mathrm{DDSM}}$. The best value for each metric is highlighted in bold. We do not measure it on FFHQ since the associated model is not trained with Dropout.

| | CIFAR-10 ($32 \times 32$) | | |
|---|---|---|---|
| MC Dropout $K = 20$ | FID $\downarrow$ | KID $\downarrow$ | $L_{\mathrm{DDSM}} \downarrow$ |
| w/o dropout | **4.83** | **0.0008** | **0.0284** (0.0006) |
| w/ dropout | 5.75 (0.07) | 0.0014 (0.0001) | 0.0286 (0.0003) |
| Arithmetic mean | 5.85 | 0.0014 | 0.0286 (0.0005) |
| Geometric mean | 5.87 | 0.0014 | 0.0286 (0.0003) |
| Dominant | 8.05 | 0.0044 | 0.0285 (0.0004) |
| Median | 5.95 | 0.0016 | 0.0285 (0.0002) |
| Alternating sampling | 8.21 | 0.0017 | - |
| Mixture of experts | 7.47 | 0.0016 | - |

negative effects of ensembling on CIFAR-10 are mitigated when it is applied over fewer timesteps. However the differences remain small overall, indicating that the results are essentially similar.

### E.4 Arithmetic mean and NLL

The model log-likelihood is computed according to the explicit formula and method given in Appendix B. We use 3 Monte Carlo samples to estimate the trace.

The metric shows a marginal downward trend with increasing $K$. (see Appendix B).

### E.5 Effect of the number of sampling steps on ensembling performance

DDIM enables sampling with a number of diffusion steps that is 10 to 100 times smaller than that used during DDPM training (e.g. 1000), while still maintaining satisfactory image quality (Song et al., 2020) as long as the number of steps is not too small. In Figure 10, we use numbers lower than 100, which we initially selected as it provided the best trade-off between speed and quality.

Table 6: We complement results from Table 3 by evaluating ensemble performance for $K = 5$ models trained on disjoint subsets of CIFAR-10 in terms of FID, KID, and $L_{\text{DDSM}}$. Each model is trained on two classes. The arithmetic mean produces catastrophic results: generated images are extremely noisy. This is because we are combining models of fundamentally different natures. Moreover, if we consider that it approximates a PoE, such a failure is unsurprising, as it essentially attempts to sample from an empty support. Mixture of experts performs better, as it avoids this issue, but the models remain too domain-specific to fully exploit diversity and reach high performance.

| | CIFAR-10 ($32 \times 32$) | | |
| Subsets w/o intersection $K = 5$ | FID $\downarrow$ | KID $\downarrow$ | $L_{\text{DDSM}} \downarrow$ |
|---|---|---|---|
| Individual model DE | **4.79** (0.08) | **0.001** (0.00) | **0.0284** (0.0005) |
| Individual model | 65.42 (7.66) | 0.037 (0.00) | 0.0381 (0.0008) |
| Arithmetic mean | 90.16 | 0.072 | 0.0345 (0.0006) |
| Mixture of experts | 12.05 | 0.007 | - |

Table 7: We evaluate FID and KID when using a Deep Ensemble where arithmetic averaging of the scores is applied only during the first one-third of the 1000 diffusion timesteps. After this early aggregation phase, sampling proceeds with a single model for the remaining steps. This experiment is repeated five times, each time using a different model from the ensemble.

| | CIFAR-10 ($32 \times 32$) | |
| Early ensembling $K = 5$. | FID $\downarrow$ | KID $\downarrow$ |
|---|---|---|
| Individual model DE | **4.79** (0.08) | **0.001** (0.00) |
| Arithmetic mean DE | 4.98 | 0.001 |
| Arithmetic mean | 4.83 (0.08) | 0.001 (0.00) |

# F  Tables and plots for Random Forests

In this section, we provide additional tables on ensemble strategies to complement the part on Random Forest (Section 5.2.4). We vary datasets and metrics. Our conclusion is that **Dominant feature** dominates in all settings, and Section 5.2.4 and fig. 11 explain why.

## F.1  Wasserstein distances on additional datasets

We present detailed results on Iris dataset and three additional ones larger than Iris (Tables 8 to 11). We still perform three train/test splits and report averaged results. The Dominant feature aggregation scheme shows in any case the best performance on generated samples. For other methods, increasing $K$ does not significantly improve generation quality.

Table 8: Wasserstein distance $W_{\text{test}}$ for different aggregation methods as a function of the number of trees $K$. We perform three different train/test splits for each experience.

| | $W_{\text{test}}$ on Iris dataset | | | | |
| Number of trees $K$ | Arithmetic | Geometric | Dominant | Median | Alternating |
|---|---|---|---|---|---|
| 1 | 0.96 | 0.96 | 0.96 | 0.96 | 0.96 |
| 25 | 0.94 | 0.94 | 0.50 | 0.98 | 0.94 |
| 50 | 0.94 | 0.94 | 0.49 | 0.98 | 0.96 |
| 100 | 0.94[*] | 0.94 | 0.46 | 0.99 | 0.95 |
| 500 | 0.94 | 0.94 | 0.46 | 0.99 | 0.95 |
| 1000 | 0.94 | 0.94 | 0.46 | 0.99 | 0.96 |

[*]Value reported by Jolicoeur-Martineau et al. (2024).

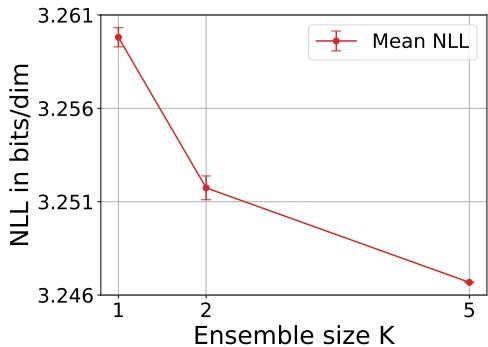

Figure 9: Mean NLL computed on CIFAR-10 validation set, with the ensemble size $K$ in the x-axis. The uncertainty bands are computed exactly in the same way as FID (Appendix A).

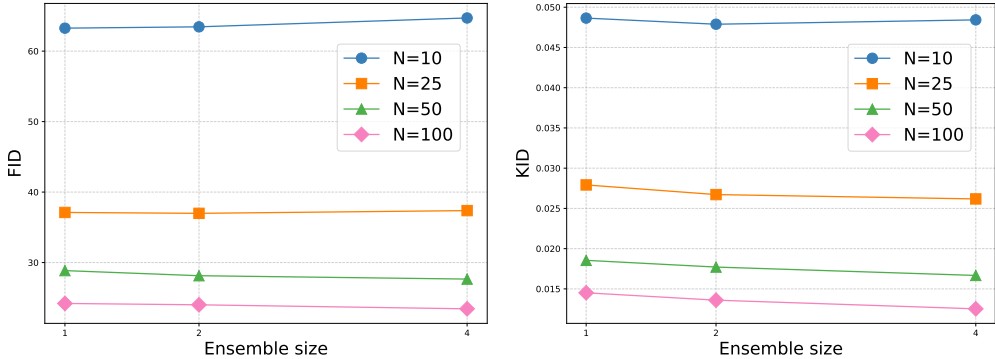

Figure 10: Effect of the number of diffusion steps at inference on perceptual quality on FFHQ for different ensemble sizes $K$. Both FID and KID remain nearly constant with respect to $K$, showing that ensembling does not mitigate discretization errors.

## F.2 Coverage and efficiency

We measure coverage and efficiency (see Appendix A for short descriptions) on the Iris dataset. We use the same range for number of trees as above, and aggregation schemes are also unchanged. **Dominant feature** performs in comparison to other methods in terms of diversity (Table 12) and in post-generation classification (Table 13). This result aligns with Table 8 and gives further credence to it.

Table 12: Coverage score $cov_{\text{test}}$ for different aggregation methods as a function of the number of trees $K$.

| Number of trees $K$ | $cov_{\text{test}}$ on Iris dataset | | | | |
| | Arithmetic | Geometric | Dominant | Median | Random select |
|---|---|---|---|---|---|
| 1 | 0.31 | 0.31 | 0.31 | 0.31 | 0.31 |
| 25 | 0.35 | 0.32 | 0.81 | 0.27 | 0.36 |
| 50 | 0.35 | 0.34 | 0.81 | 0.27 | 0.32 |
| 100 | 0.34 | 0.34 | 0.82 | 0.28 | 0.29 |
| 500 | 0.35 | 0.35 | 0.83 | 0.30 | 0.33 |
| 1000 | 0.34 | 0.35 | 0.85 | 0.29 | 0.31 |

Table 9: Wasserstein distance $W_{\text{test}}$ for different aggregation methods as a function of the number of trees $K$ on the Airfoil Self Noise dataset ($n = 1503$, $d = 5$, Brooks et al., 1989).

| Number of trees $K$ | $W_{\text{test}}$ on Airfoil Self Noise dataset | | | | |
| | Arithmetic | Geometric | Dominant | Median | Alternating |
|---|---|---|---|---|---|
| 1 | 0.88 | 0.88 | 0.88 | 0.88 | 0.88 |
| 25 | 0.88 | 0.89 | 0.58 | 0.92 | 0.88 |
| 50 | 0.88 | 0.89 | 0.55 | 0.92 | 0.88 |
| 100 | 0.88 | 0.89 | 0.54 | 0.92 | 0.88 |
| 500 | 0.88 | 0.89 | 0.53 | 0.92 | 0.88 |
| 1000 | 0.88 | 0.89 | 0.52 | 0.92 | 0.88 |

Table 10: Wasserstein distance $W_{\text{test}}$ for different aggregation methods as a function of the number of trees $K$ on the Wine Quality White dataset ($n = 4898$, $d = 11$, Cortez et al., 2009).

| Number of trees $K$ | $W_{\text{test}}$ on Wine Quality White dataset | | | | |
| | Arithmetic | Geometric | Dominant | Median | Alternating |
|---|---|---|---|---|---|
| 1 | 3.71 | 3.71 | 3.71 | 3.71 | 3.71 |
| 25 | 3.69 | 3.90 | 2.69 | 3.60 | 3.71 |
| 50 | 3.70 | 3.91 | 2.62 | 3.59 | 3.71 |
| 100 | 3.69 | 3.90 | 2.55 | 3.59 | 3.71 |
| 500 | 3.69 | 3.90 | 2.43 | 3.58 | 3.70 |
| 1000 | 3.69 | 3.90 | 2.38 | 3.58 | 3.70 |

Table 13: F1 score $F1_{\text{test}}$ for different aggregation methods as a function of the number of trees $K$.

| Number of trees $K$ | $F1_{\text{test}}$ on Iris dataset | | | | |
| | Arithmetic | Geometric | Dominant | Median | Random select |
|---|---|---|---|---|---|
| 1 | 0.75 | 0.75 | 0.75 | 0.75 | 0.75 |
| 25 | 0.78 | 0.79 | 0.87 | 0.76 | 0.79 |
| 50 | 0.78 | 0.80 | 0.86 | 0.78 | 0.73 |
| 100 | 0.78 | 0.78 | 0.88 | 0.77 | 0.75 |
| 500 | 0.77 | 0.78 | 0.90 | 0.77 | 0.76 |
| 1000 | 0.79 | 0.78 | 0.88 | 0.77 | 0.74 |

Table 11: Wasserstein distance $W_{\text{test}}$ for different aggregation methods as a function of the number of trees $K$ on the Wine Quality Red dataset ($n = 1599$, $d = 10$, Cortez et al. (2009)).

| | $W_{\text{test}}$ on Wine Quality Red dataset | | | | |
|---|---|---|---|---|---|
| Number of trees $K$ | Arithmetic | Geometric | Dominant | Median | Alternating sampling |
| 1 | 3.23 | 3.23 | 3.23 | 3.23 | 3.23 |
| 25 | 3.23 | 3.39 | 2.18 | 3.11 | 3.25 |
| 50 | 3.23 | 3.39 | 2.09 | 3.10 | 3.25 |
| 100 | 3.23 | 3.39 | 2.01 | 3.10 | 3.25 |
| 500 | 3.23 | 3.38 | 1.85 | 3.09 | 3.25 |
| 1000 | 3.23 | 3.38 | 1.80 | 3.09 | 3.25 |

### F.3 Noise underestimation

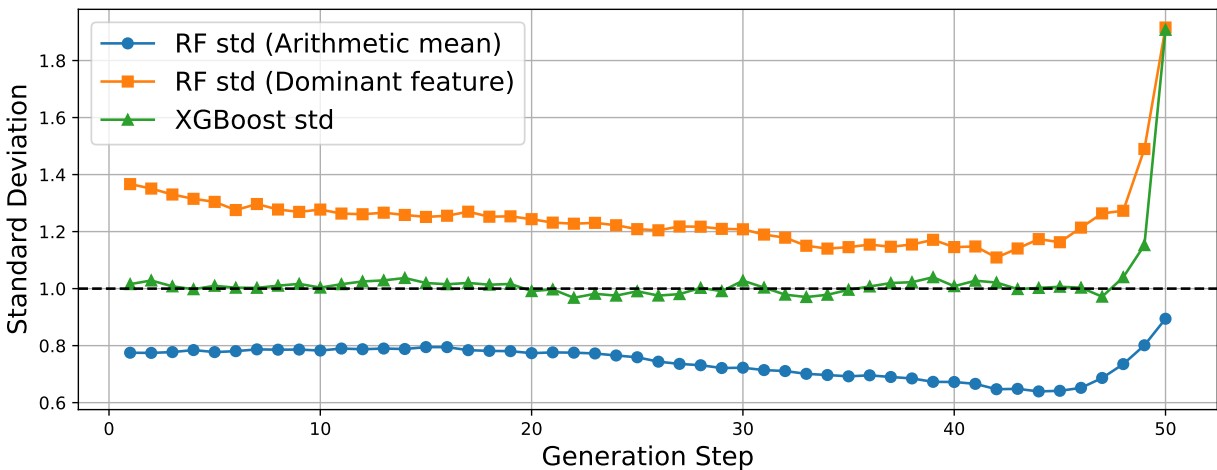

Figure 11: We compare the overall standard deviations of the two-dimensional predicted scores on the *Iris* dataset. Specifically, we evaluate the Arithmetic Mean and Dominant Component methods against XGBoost, which is the best-performing score model according to Jolicoeur-Martineau et al. (2024). To ensure a fair underfitting analysis, all models were trained with the same maximum tree depth of 7, which is sufficient for this dataset. The Dominant Component method yields higher predicted noise magnitudes than the Arithmetic Mean, closely matching XGBoost's standard deviation at the final steps.

## G Weight scaling at initialization

To the best of our knowledge, no prior work has examined the benefits of initializing a score network with weights scaled close to zero. Yet, as we will demonstrate, this choice positively influences both training behavior and final performance. In this section, we review related methods and illustrate how this initialization affects convergence on $L_{\text{DDSM}}$ and FID.

### G.1 Near-zero output initialization on score models

In these architectures, whether the network predicts the score (Song et al., 2021b) or the noise (Ho et al., 2020; Nichol & Dhariwal, 2021), the final layer's weights are by design initialized near zero. For example, Nichol & Dhariwal (2021) enforce this via their `zero_module` implementation on the final convolutional layer, while Song et al. (2021b) apply Xavier uniform initialization (Glorot & Bengio, 2010) with a very

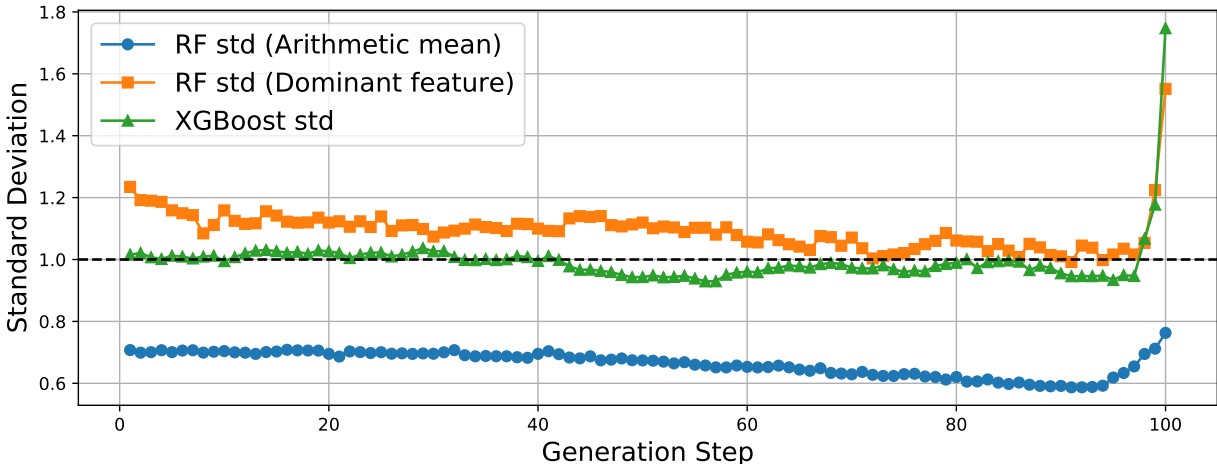

Figure 12: We compare the overall standard deviations on more diffusion steps (100). Increasing the number of diffusion steps does not allow Random Forest to reach the score estimation quality of the stronger models.

small scale parameter (e.g., $10^{-10}$), effectively constraining the network's output to be a zero tensor at the start of training. In this experiment we upscale $\lambda$ to one.

### G.2 Related works on general contexts

ControlNet (Zhang et al., 2023) almost applies this principle but in the context of fine-tuning, by initializing newly added convolutional layers to zero allowing the model to incorporate conditioning without immediately altering the behavior of the pre-trained diffusion backbone. A similar effect has been studied in the context of deep residual networks, where De & Smith (2020) show that Batch Normalization facilitates training by pushing residual blocks toward the identity function. However, subsequent work demonstrated that BatchNorm is not strictly necessary, as a simple modification to the initialization (such as setting the residual branch to zero at the start) suffices to achieve similar trainability (Zhang et al., 2019). This suggests that initializing the final layer near zero in diffusion models may play a comparable role, ensuring smooth early training dynamics without requiring explicit normalization layers.

### G.3 Effect of weight scaling on training

We verify how scaling down $\lambda$ helps stabilizing training using the DDPM++ model on CIFAR-10 (same settings as Section 5).

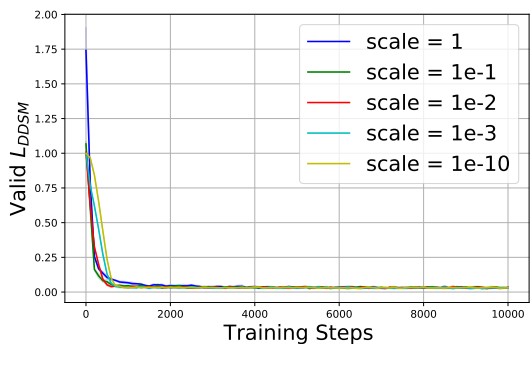
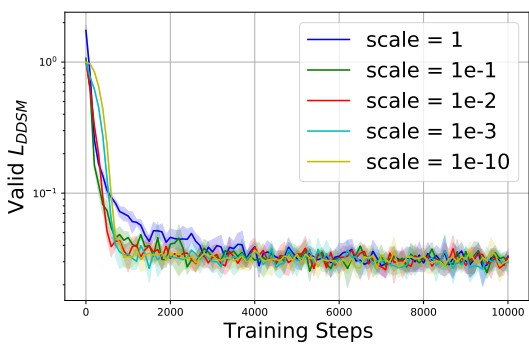

(a) Losses at normal scale.                    (b) Losses in log scale.

Figure 13: $L_{\text{DDSM}}$ evaluated on the validation set of CIFAR-10 over the course of training up to 200k iterations. We evaluate different scales $\lambda$ ($10^{-j}$ for $j \in \{0, 1, 2, 3, 10\}$). We observe that compared to using a scale of $\lambda = 1$, reducing the scale below one brings the average loss closer to 1 at the first step and results in faster convergence during the early stages of training. Despite this, the losses quickly stabilize around similar values regardless of the initial scale.

## G.4 Effect of weight scaling on FID

We show the positive effect of near-zero scaling of initialization from the DDPM++ architecture on sample quality (same settings as Section 5).

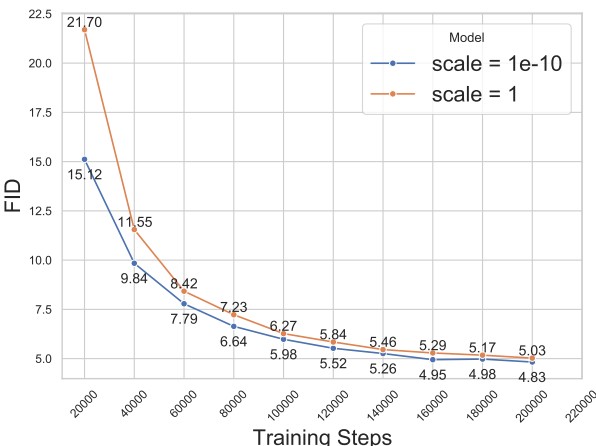

Figure 14: FID-10k on CIFAR-10 evaluated over the course of training on the two extreme cases of $\lambda$ up to 200k iterations. While both models improve their FID-10k score over the course of training, the model initialized with a scale of $10^{-10}$ consistently outperforms the one with scale 1, which fails to reach the same level of performance before 200k steps (the gap consistently stays above 0.2).

## G.5 Effect of weight scaling on predictive diversity

We show in Figure 4 that increasing the weight scale at initialization enhances post-training diversity.

We measure the predictive (or functional) diversity, which is the diversity in the output space. Since each model is trained by minimizing the MSE of scores or noises depending on the framework, we measure the predictive diversity by adopting the metric associated to the squared loss and arithmetic mean combiner from Wood et al. (2023). Given $K$ models $\boldsymbol{s}_{\boldsymbol{\theta}}^{(1)}, \ldots, \boldsymbol{s}_{\boldsymbol{\theta}}^{(K)}$ taking a data point $\mathbf{x}$ and a timestep $t$ as input and

producing an output in the same space as $\mathbf{x}$, we define *predictive diversity* as the variance of the predictions and write it as

$$D^{(K)}(\boldsymbol{s}_{\boldsymbol{\theta}}^{(1)}, \ldots, \boldsymbol{s}_{\boldsymbol{\theta}}^{(K)}) = \mathbb{E}_{(\mathbf{x},t)}\left[\frac{1}{K}\sum_{k=1}^{K}(\boldsymbol{s}_{\boldsymbol{\theta}}^{(k)}(\mathbf{x},t) - \overline{\boldsymbol{s}}_{\boldsymbol{\theta}}^{(K)}(\mathbf{x},t))^2\right] \tag{98}$$

where $\overline{\boldsymbol{s}}_{\boldsymbol{\theta}}^{(K)}(\mathbf{x},t) = \frac{1}{K}\sum_{k=1}^{K}\boldsymbol{s}_{\boldsymbol{\theta}}^{(k)}(\mathbf{x},t)$ is the combination of the predictors, and the average of squared differences is calculated per image rather than per pixel.

## H    Note on FID and related metrics

Automated perceptual metrics remain standard and widely used tools for comparing generative models. They provide an essential, reproducible baseline for evaluation, and are often indispensable when large-scale human studies are impractical or unavailable. However, if typically we take FID which is the most popular one, it is frequently used without a thorough assessment of its limitations and its relevance as a proxy for visual quality has been repeatedly questioned in the literature (Stein et al., 2023; Jayasumana et al., 2024; Karras et al., 2020; Borji, 2022; Morozov et al., 2021). In particular, Stein et al. (2023) highlight cases where FID (and network feature-based metrics in general) correlates poorly with human judgment, notably showing that FID underestimates the quality of diffusion models on FFHQ compared to human perception. Another point is that FID, IS and KID rely on features extracted from Inception networks pre-trained on ImageNet. Yet several works showed limits of their use on datasets that differ significantly from ImageNet, such as human faces (Borji, 2022; Kynkäänniemi et al., 2023). For instance, Kynkäänniemi et al. (2023) show that FID is sensitive to the alignment between the generated and ImageNet class distributions, which can result in misleading evaluations particularly on datasets like FFHQ, where the label distribution deviates substantially from that of ImageNet.

