# OpenReview forum: "When Are Two Scores Better Than One? Investigating Ensembles of Diffusion Models"
_TMLR — Accepted by TMLR_

### Review · Reviewer_22sG · 2025-08-28

**Summary Of Contributions:**

This paper presents a comprehensive investigation into the effectiveness of ensembling techniques for unconditional score-based diffusion models. The authors explore whether combining multiple models provides similar benefits in the context of generative modeling.
From my point of view, the primary contributions are:

Empirical Evaluation: The authors conduct extensive experiments across multiple datasets (CIFAR-10, FFHQ, tabular data), model architectures (U-Nets, Random Forests), and ensemble methods (Deep Ensembles, MC Dropout). They test a variety of aggregation rules for combining model outputs, including arithmetic mean, median, and more heuristic approaches.

Key Finding of a Disconnect: The central finding is that while ensembling consistently reduces the score-matching loss (L
DDSM) and improves model likelihood, it fails to reliably improve perceptual quality metrics like Fréchet Inception Distance (FID) or Kernel Inception Distance (KID). The ensemble often performs better than the average individual model but rarely surpasses the single best-performing model in the ensemble.

Theoretical Insights: The paper provides theoretical justification for some of the empirical results. It formally shows why averaging scores reduces the expected L_{DDSM} loss. Crucially, it also demonstrates with a clear counterexample that the operations of diffusion (adding noise) and model composition (e.g., via a Product-of-Experts) do not commute (Proposition 4.2). This clarifies a common heuristic used in model guidance and composition.

**Additional Comments:**

No.

**Audience:**

Yes

**Audience Explanation:**

It is a nice paper, and I think that there will be an audience of TMLR interested in it.

Relevance to Diffusion Model Practitioners: Diffusion models are a central topic in modern machine learning. Ensembling is a fundamental and widely used technique. This paper addresses the natural and important question of whether these two concepts can be combined effectively. The largely negative result is itself a significant contribution, as it can save other researchers significant time and effort by showing that the most straightforward application of ensembling is not fruitful for improving sample quality.

Implications for Generative Model Evaluation: The paper's investigation into why ensembling fails highlights the critical disconnect between common training objectives (like score matching) and popular perceptual metrics (like FID). This is a broader issue relevant to the entire field of generative modeling. It encourages the community to think more deeply about how we train and evaluate these models, a topic of ongoing research and debate.

Theoretical Value: The theoretical analysis of score composition, particularly the proof that diffusion and PoE do not commute, has implications beyond ensembling. It provides a formal basis for understanding why techniques like classifier-free guidance are heuristics rather than exact methods for sampling from a target product distribution. This will be of interest to researchers working on the theory and foundations of diffusion models

**Broader Impact Concerns:**

No.

**Claims And Evidence:**

Yes

**Claims Explanation:**

**Claim**: Ensembling improves score loss but not perceptual quality. This is the paper's core claim and is strongly supported. Figure 3e clearly shows a monotonic decrease in the $L _{DDSM}$ loss as the ensemble size (K) increases for CIFAR-10. In contrast, Figures 3a-d and the visual example in Figure 1 show that FID and KID do not consistently improve and that the ensemble fails to beat the strongest individual model. Table 1 provides further quantitative evidence, where on FFHQ, the "Arithmetic mean" ensemble achieves an FID of 23.44, which is better than the average individual model (24.20) but worse than the best individual model (21.7).

**Claim**: Alternative ensemble strategies offer limited improvement. The authors comprehensively test numerous aggregation rules in Table 1. Most methods that aggregate scores at each step (mean, median, geometric mean) show marginal or no improvement over the simple average. The "Mixture of experts" strategy does show a notable improvement on FFHQ , but the authors correctly identify that this is not a true ensemble in the traditional sense, as no aggregation occurs within the generation of a single sample. Their investigation into MC Dropout (Table 5) and diversity-promoting strategies (Tables 2 and 3)  further strengthens the conclusion that simple ensembling is not a silver bullet.

**Claim**: The failure is due to a disconnect between the training objective and perceptual metrics. This explanation is well-argued and supported. Figure 6 provides compelling evidence: Figures 6a and 6b show that during a single model's training, the validation $L_DDSM$
​stagnates while FID continues to improve, demonstrating their different behaviors. Furthermore, Figures 6c and 6d brilliantly illustrate that small perturbations added to the score function cause large, chaotic swings in FID and KID, highlighting the sensitivity of these perceptual metrics in a way that the smooth, averaged $L_2$ score loss is not.

Theoretical Claims: The proofs provided in the appendix are clear and appear correct. Proposition 4.1 is a straightforward but important application of Jensen's inequality to the diffusion context. Proposition 4.2, which proves the non-commutativity of diffusion and Product-of-Experts with a simple Gaussian case, is a valuable and elegant theoretical contribution that formalizes a subtlety often overlooked in guidance-based methods

**Requested Changes:**

**Emphasize the "Mixture of Experts" Finding**: In Table 1, the "Mixture of experts" method is the only one that clearly outperforms the best single model on the high-resolution FFHQ dataset (FID 20.36 vs. 21.7). While you correctly note this isn't a "true" ensemble because scores aren't averaged within a sample's generation, this is still a very interesting result. It implies that the diversity captured by independently trained models is valuable, but the act of averaging their scores is what is detrimental to perceptual quality. This point could be highlighted more strongly in the discussion and conclusion. It suggests that future work should focus on methods that can leverage inter-model diversity without the "blurring" effect that simple averaging seems to cause.

---

> ### Author Response · Authors · 2025-10-29
> **Answer to reviewer 22sG**
>
> Many thanks for the detailed and thoughtful comments, as well as for the valuable suggestion (i) regarding the Mixture of Experts. We agree that this finding deserves to be emphasized, and highlighting it indeed strengthens the paper.
>
> > Emphasize the "Mixture of Experts" Finding.
>
> We agree that this is an important and insightful observation that deserves to be highlighted. We therefore revised the the contributions in the introduction (page 2) to better emphasize this result, and we added a note in the conclusion (page 13) discussing its significance and the conditions under which such complementary models can be beneficial. That said, while we fully acknowledge this notable exception, we also discuss in the conclusion the benefit–cost trade-off, noting that the computational cost of ensembling remains substantial and that our overall assessment that the benefit–cost ratio is generally unfavorable still holds.

---

### Review · Reviewer_wo7R · 2025-09-07

**Summary Of Contributions:**

The paper studies the effect of the ensembling techniques on the diffusion models. Experiments on different tasks and models show that it improved the score matching objective but does not help the perception metrics (e.g. FID score).

**Additional Comments:**

Minor corrections:
1. Figure 3 (CIFAR): Why loss have the result for up to 10 components, but FID and KID for up to 5
2. Typo (Section 5.3.1): “the optimized objective differs from **te** one of the perceptual metric of interest”

**Audience:**

No

**Audience Explanation:**

I think that the community, especially the practitioners, would be interested to see how useful (or not) the ensembling techniques are for diffusion models.
However, I think for the paper to be insightful, a more diffusion-specific experiments are required. For example:
1. Do we need to apply ensembling to all the timestamps? Or will it have a different effect? E.g. closer to the “data space” the model is generating high-frequency components that are important for FID score. Maybe averaging is harmful there but can help on the earlier stages
2. Can we reduce the amount of sampling steps but keep the performance comparable using ensembling? That is, can the averaging smooth out the larger discretization error when we use fewer diffusion steps?

**Claims And Evidence:**

Yes

**Claims Explanation:**

Experiments on different dataset and models demonstrate the effect of ensembling on the model performance. Also different aggregation methods are explored. However, I have questions/comments about some of the contributions stated in the introduction:

1. ”We also provide theoretical insights on its [ensembling]  effect on the training objective error”. I would not call this a theoretical insight as it directly follows from the definitions.
2.  “[We] contextualize ensembling within other contexts”. I do not really understand what is meant by this. I would be happy to hear the authors’ clarifications.
3. “addressing the mismatch between the score matching loss and perceptual image quality metrics”. From this formulation it seems that the authors address the problem of the mismatch between the loss and the metric, which is not the case. The mismatch is indeed empirically demonstrated, but I would say that this is a widely known fact for diffusion models. Also, not obvious, why addressing the mismatch would help the ensembling.

**Requested Changes:**

1. Please clarify / correct the contributions
2. Please add a more insightful experiment to demonstrate why DE do not work that well or how it can be useful specifically for diffusion models (see specific suggestions above)

---

> ### Author Response · Authors · 2025-10-29
> **Answer to reviewer wo7R**
>
> Many thanks for the detailed and constructive feedback. Your comments were very helpful in identifying points that required clarification and led us to make several improvements on the paper, particularly in the experimental section (suggestion (iv) in the general comment).
>
> > ”We also provide theoretical insights on its [ensembling] effect on the training objective error”. I would not call this a theoretical insight as it directly follows from the definitions.
>
> We understand your point that this result is straightforward from the definitions, but we believe it is still useful to make the result explicit and formal within the diffusion context. This result clarifies an important conceptual point within this framework.
>
> Thanks to your comment, we revisited this result and realized that a stronger statement could be made. In the revised version, we replaced Proposition 4.1 (page 7) with a less straightforward and more general result establishing the monotonicity of the training objective with respect to the ensemble size.
>
>
> >What does "contextualize ensembling within other contexts" mean?
>
> We relate ensembling, seen as a form of model composition $\propto p^{(1)}(\mathbf{x})^{\frac{1}{2}} p^{(2)}(\mathbf{x})^{\frac{1}{2}}$, to mechanisms from different contexts that also combine multiple diffusion models. In Section 3.3, we emphasize how averaging scores mathematically resembles model combination schemes such as guidance (in which the target is a product involving conditional distributions). We make this connection to show that our theoretical results, particularly the non-commutativity of Product-of-Experts with diffusion, are not limited to ensembling but apply to these broader contexts as well.
>
> > The authors do not address the problem of the mismatch between the loss and the metric.
>
> We agree that our wording “addressing the mismatch” was misleading, as our work does not propose a method to resolve this discrepancy. Our intention was rather to analyze and illustrate how this well-known mismatch between the score matching loss and perceptual quality metrics manifests in the context of ensembling. We have revised the phrasing at the end of the introduction (page 2) to better reflect this clarification. We also rewrited the analysis of Figure 7 (page 12) and modified it by replacing the KID evolution w.r.t noise levels by $L_{\text{DDSM}}$. on 7b.
>
> > Do we need to apply ensembling to all the timestamps?  Maybe averaging can help on the earlier stages.
>
> Thanks for suggesting this, we agree that the impact of sampling steps was overlooked. We consequently added in Appendix (Table 7, page 35) an additional experiment where ensembling is applied only during the first third of the sampling steps. The results show no improvement compared to individual models. However, the method performs better on average than averaging across all timesteps, suggesting that reducing the number of averaging operations mitigates the blurring effect that could be associated with score averaging. Thank you for this valuable suggestion, which helped us strengthen our observations and better understand the ensembling effect.
>
> > Can we reduce the amount of sampling steps but keep the performance comparable using ensembling?
>
> We added Figure 10 in Appendix (page 36), which reports the FID as a function of both the ensemble size $K$ and the number of diffusion steps $N_{\text{steps}}$. The results show that for all tested values of $N_{\text{steps}}$ (below 100), the FID curves remain essentially flat as $K$ increases. This indicates that ensembling cannot compensate here for the discretization errors that arise when using fewer sampling steps.
>
> > Why loss have the result for up to 10 components, but FID and KID for up to 5?
>
> Deep Ensembles are very costly in particular in the diffusion model context, so to keep the experimental cost manageable we limited FID and KID to ensembles up to $K = 5$, whereas the loss curves could be extended to $K = 10$ at a much lower cost.
>
> > Typo (Section 5.3.1): “the optimized objective differs from te one of the perceptual metric of interest”
>
> Thanks for pointing these out. We corrected these ones in the new version.

---

### Review · Reviewer_9hb2 · 2025-10-14

**Summary Of Contributions:**

This manuscript studies whether ensembling improves generation in unconditional score-based diffusion models. The authors report small but consistent gains in score-matching loss and likelihood, yet marginal to no improvements on perceptual metrics (FID/KID) with Deep Ensemble models on image datasets. They also provide theoretical analysis to provide insights as to why averaging scores reduces the DDSM objective and motivate why ensembling for score estimation is challenging. These findings are complimented with several practical insights and benchmarks on CIFAR-10 and FFHQ, as well as some tabular datasets such as Iris and Wine Color which involve Random Forest models for score estimation.

**Additional Comments:**

Not applicable.

**Audience:**

Yes

**Audience Explanation:**

Ensembling is a key tool to improve ML performances in several areas of research. Investigating its effects in the context of diffusion models bears interest for the ML community in general, both from theoretical and practical perspective.

**Broader Impact Concerns:**

Not applicable.

**Claims And Evidence:**

No

**Claims Explanation:**

The core findings on score loss are accurate and clearly shown: averaging scores can lower the loss and improve likelihood as mathematically proved by the authors. Yet, the generality implied in the abstract that ensembling yields marginal to no improvement for diffusion models is not convincingly established beyond specific choices.

In particular, the class of Deep Ensemble models considered in this work shows major limitations that are not mentioned nor explored by the authors. Then, while the lack of data types (mostly images) prevents generalizability across domains, the evaluation design remains often unsatisfactory due to its lack of proper statistical validation through multiple runs. Finally, the paper itself shows a counter-argument to its major claim on tabular data, indicating that "Dominant" aggregation for Random Forest improves score estimation and substantially helps, ultimately limiting the scope of the original takeaway to Deep Ensemble methods (with same architecture) on image datasets at most.

**Requested Changes:**

**Major weaknesses:**
- Empirical evaluation is dominated by image datasets and score-averaging baselines of Deep Ensemble (DE) models with the same architecture. Two tabular experiments use Random Forest-based scores, without intersecting with deep ensembling, despite different behaviors being observed (no gain in the DE ensembling, while RF shows Domination ensembling gains). As a result, conclusions are mostly retained for unconditional image generation with specific DE methods using score aggregation. The scope of the contribution needs to be backed-up by further experiments, or narrowed down to a precise set of practices.
- Ensemble variants largely keep architecture and weights fixed across members (independent U-Nets with identical design; MC-Dropout; class-subset specialization; weight-scale changes). Heterogeneous ensembles like mixed architectures (U-Net, CNN, etc.) are not explored and could affect conclusions.
- Claims about lack of perceptual gains should be carefully scoped: Mixture-of-Experts outperforms the best individual model on FFHQ in Table~1, and on tabular data the Dominant aggregation improves Wasserstein distance. These are notable exceptions that the narrative could emphasize to avoid an overly broad takeaway.
- General claims relying on experiments need more grounding asserted by statistical significance (see specific Questions below). The limitations of these claims is often not considered, ultimately hindering the overall soundness.
- The authors do not provide a repository to reproduce their experiments, impairing proper verification of their practical findings.

**Questions and comments:**
- **p.2, Figure 1 ("Our observations show that an ensemble of five models does not achieve the highest image quality even though it beats the average one"):** Figure 1 contrasts strongest, weakest, and the ensemble, but not an "average" model in the FID-10k sense. Providing an illustration for the median/mean-FID model model would help supporting the claim. Besides, adding the distribution of the FID-10k scores across the ensemble method in Appendix would help visualize the discrepancies between the models of the ensemble by highlighting possible outliers.
- **p.3: "[Deep ensemble] involves constructing K models, each sharing the same neural network architecture [...] A key ingredient for its success is the diversity between the individual models":** The authors state Deep Ensemble members share the same architecture and diversity weights "naturally arises" from stochasticity of the learning process; later they cite Xu et al. (2024) noting convergence to similar solutions in diffusion models sharing the same architecture, contradicting their claim p.5: "we could expect models trained on different seeds to converge to noticeably different solutions". It seems mandatory to reconcile these points to substantiate the claim, either by testing or evaluating diversity (as in Appendix G.5) from learned weighting across members. Adding heterogeneous architectures to compare the diversity against same parameterization ensembles would be interesting as well. Alternatively the claim of diversity from the depicted ensembles should be tempered and the scope of the contribution re-assessed.
- **Equation (6) and guidance equivalence (p. 6–8):** The manuscript rightly notes that Eq. (6) and related equalities do not always hold in the context of diffusion models, then mentions MCMC/Feynman–Kac correctors. Appendix C.4 also mentions these approaches p.27. Yet, it is unclear whether these correctors were used in the experiments to properly sample from the diffusion. Please clarify whether any correctors were used in experiments; if not, the scope of the contribution should be specified.
- **Figure 3 presentation (p. 8):** The caption is not informative, axis labeling is hard to connect across subplots, and uncertainty depiction is inconsistent (only LDDSM shows "$\pm$ Std Dev"). Standardizing axis labels/legends, defining the uncertainty bands, and harmonizing the $K$ range across (a)–(e) would improve readability. Note that (e) explores up to $K=10$ while others stop earlier, even though interpretations would benefit from more points, why is that?
- **Table 1 caption (p. 9):** The caption is incomplete as some elements of the table are not described. Clarify which entries include standard deviations (parentheses appear for individual models but not all ensembles, probably because the experiment was not repeated to assess variability). Reporting variability and the number of training runs for all entries would improve substantiate the claim.
- **About weight scaling with $\lambda$: "This suggests that increasing predictive diversity alone is insufficient to enhance ensemble performance in diffusion models." (p. 10):** Given Figure 11 provides the core analysis of this claim, the authors should consider bringing it into the body of the manuscript. Besides, the authors are encouraged to discuss possible trade-offs  and limitations of this claim (e.g., when does diversity help/harm).
- **"We observe that ensembles built from weaker models do yield bigger improvements in FID, while gains diminish and even turn negative as the base models become stronger" (p. 11):** Due to the lack of repetitions to estimate $\Delta$ at each iteration, Figure 4 might be insufficient in its current state to support this statement. It would be mandatory to assess the variability of $\Delta$ to substantiate the claim and assert a pattern. Similar results with other values of $K$ would be expected to show a convincing trend in $\Delta$. Also, the definition of $\Delta$ in the Caption of Figure 4 should be provided out of the descriptor (a) as it is also used in (b), and plotted confidence regions need to be defined.
- **Figure 5 (p. 11):** The authors mention a reference models from the study of Jolicoeur-Martineau et al. (2024) in the context of this experiment. Adding the reference performance onto Figure 5 would help situating the improvements made by the Dominant aggregation. Also, adding explicit $x$/$y$ labels would improve readability.
- **Figure 8 interpretation (p. 35):** The figure shows standard deviations of predicted scores (Arithmetic vs. Dominant vs. XGBoost). Please clarify the relationship between under/overestimation of noise magnitude and Wasserstein performance, and whether increasing the amount of denoising steps or alternative schedulers would change the conclusion in the underestimation regime.
- **"It may be caused by an underfitting effect coming from decision trees" (p. 11):** This claim requires further supporting evidence, as it does not align with typical behaviors of Random Forest. Appendix A.2 does not list tree depths, which can be relating to under/overfitting in some instances.
- **Figure 6 and "the disconnect between what diffusion models are trained to optimize (score matching) and how they are typically evaluated explains why ensembling struggles to diminish image quality metrics." (p. 12):** The authors are encouraged to add confidence intervals over multiple training seeds and a  $\log y$-scale for readability where appropriate. The sensitivity plots are central to the argument that FID reacts chaotically to small score perturbations; thus additional repetitions would strengthen this point. The validity of the aforementioned claim appears shaky without mathematical consolidations or improved benchmarks.
- **Sampling procedure missing (Appendix A):** It would help to consolidate sampling details: solver (SDE/ODE or DDIM), step counts, $\eta$ or SNR schedules, guidance/temperature (if any), usage of correctors, and seeds for each experiment. Some of this is dispersed in the preliminaries but not fully specified per experiment.
- **Appendix C.4 (proof clarity):** This counterexample is a key result of the manuscript. The proof would read more smoothly with proper declaration of fixed quantities and a lemma-first structure for intermediate steps. Some notational tightenings would also be appreciated to make the proof more rigorous.
- **Appendix C.4 scope/title:** This appendix also includes remarks on concept-conditional modeling and linear inverse problems. The authors should consider retitling or splitting into subsections so readers can locate these topics easily. Also, it is unclear why linear inverse problems are mentioned. The section should either be more clearly involved in the manuscript or removed.
- **Metric formatting in tables:** Reporting KID and LDDSM with a consistent scientific-notation factor (e.g., $\times 10^k$) would improve readability across tables.
- **Appendix E.2, Figure 7:** The uncertainty band dominates the plot, making trends hard to interpret. The authors should consider adjusting scale, adding more runs, or plotting relative changes to improve readability. As is, the figure bears limited utility.
- **Cited work:** Several citations are preprints. Where possible, the authors should prefer the latest peer-reviewed versions and consider replacing some instances with recently published work on ensemble guidance/density composition.

---

> ### Author Response · Authors · 2025-10-29
> **Answer to reviewer 9hb2 (1/3)**
>
> Many thanks for the detailed and constructive feedback. Your comments were very helpful in improving the clarity and scope of the paper. We particularly appreciate your remarks on the exceptions (such as the MoE and Dominant) as well as your insightful comments regarding the variability analysis and statistical significance. These observations helped us refine our discussion, clarify our methodology, and better highlight the nuances of our results.
>
> **On the overall message**
>
> > Claims about lack of perceptual gains should be carefully scoped: Mixture-of-Experts outperforms the best individual model on FFHQ in Table~1, and on tabular data the Dominant aggregation improves Wasserstein distance
>
> We fully agree with this suggestion. We now emphasize these exceptions more clearly in the revised version. In particular, we mentionned that the MoE performs better on FFHQ in a specific setting where the models are sufficiently diverse to be truly complementary, which we now highlight in the contribution paragraph with a new bullet point, and in the conclusion to stress the importance of diversity. As for the Dominant aggregation, we note that its improvement on tabular data mainly stems from mitigating an underlying underfitting issue rather than from ensembling itself (similar gains could be obtained by upscaling the outputs of the Random Forest). We added this clarification in Section 5.2.4 page 9.
>
>
> **On organisation of experiments and clarification**
>
> > Adding the distribution of the FID-10k scores across the ensemble.
>
> We added FID-10k score distributions across members of the DE in Figure 1 (p2, FFHQ) and Figure 8 (p34, CIFAR-10). One important observation we emphasized (p9 & p34) is that, on CIFAR-10, the ensemble can even perform worse than the weakest individual model, likely due to a blurring effect caused by score averaging. It illustrates that ensembling can sometimes be detrimental.
>
> > Describe the sampling procedure (Appendix A, p. 19).
>
> We described in more details the sampling procedure on image generation in Appendix A (p19).
>
> > Given Figure 11 provides the core analysis of this claim, the authors should consider bringing it into the body of the manuscript. Besides, the authors are encouraged to discuss possible trade-offs and limitations of this claim (e.g., when does diversity help/harm).
>
> We moved Figure 11 (now Fig. 4 p10) into the main text to make the analysis clearer and support the discussion. As shown in Tables 2–3, diversity never improves score-averaging performance; increasing predictive diversity alone does not help. The MoE leverages diversity more effectively, though its benefit remains limited (works only in ffhq). We discuss the role of diversity for MoE in the conclusion (§2 p8).
>
> > The authors mention a reference models from the study of Jolicoeur-Martineau et al. (2024) in the context of this experiment. Adding the reference performance onto Figure 5 would help situating the improvements made by the Dominant aggregation.
>
> Thank you for pointing out this omission. We updated Figure 5 to include the reference performance. It corresponds to 100 trees ($W_{\text{test}} = 0.94$).
>
> > It may be caused by an underfitting effect coming from decision trees” (p11): This claim requires further supporting evidence, as it does not align with typical behaviors of Random Forest. Appendix A.2 does not list tree depths, which can be relating to under/overfitting in some instances.
>
> Although the result differs from typical behavior, Figure 11 clearly shows underfitting because the RF predicts noise less accurately than XGBoost (best for Wasserstein). The missing tree-depth detail was added in App A.2 (p20); **all models** including XGBoost were trained with a maximum depth of 7, consistent with the reference paper and sufficient for small datasets like Iris. Thus we think the RF’s underfitting cannot be due to limited depth. The Figure 11 caption was updated accordingly.
>
> > Clarify the relationship between under/overestimation of noise magnitude and Wasserstein performance, and whether increasing the amount of denoising steps or alternative schedulers would change the conclusion in the underestimation regime.
>
> The Wasserstein distance quantifies how close generated and test samples are. The Random Forest’s underestimated noise magnitude (compared to Dominant and XGBoost) reveals underfitting: noise (or score) is less accurately predicted along the sampling path from $t=T$ to final sample $t=0$. Since Figure 11 in Appendix (p39) is consistent with the Wasserstein results, we argue that this is the underlying reason. Your suggestion is interesting thus we increase the amount of steps in Figure 12 (p39) but without success.

---

> ### Author Response · Authors · 2025-10-29
> **Answer to reviewer 9hb2 (2/3)**
>
> **On experiments and uncertainty**
>
> > Table 1 caption (p. 9): The caption is incomplete as some elements of the table are not described. Clarify which entries include standard deviations. Figure 3 (p. 8): Uncertainty depiction is inconsistent
>
> We clarified the methodology of Table 1 in p8, addressing the previously missing explanation. As more detailed and justified in the Appendix A (p21), the procedure used to assess variability differs across metrics (FID is very costly compared to $L_{\text{DDSM}}$ in the ensemble settings) and depends on the purpose of each plot. For a maximum ensemble size $K_{\text{max}}$, FID variability for smaller $K$ is measured across all subsets (e.g., $K=2$ uses all pairs within $K_{\text{max}}$), hence no standard deviation for $K=5$. For $L_{\text{DDSM}}$, intervals come from 10 runs for all sizes.
>
> > Note that (e) explores up to $K=10$ while others stop earlier, even though interpretations would benefit from more points, why is that?
>
> The $K$ range was limited by computational constraints: larger ensembles were feasible in some cases but too costly in others. $L_{\text{DDSM}}$ remained affordable for $K=10$.
>
> > Figure 6 and "the disconnect between what diffusion models are trained to optimize (score matching) and how they are typically evaluated explains why ensembling struggles to diminish image quality metrics." (p. 12): The authors are encouraged to add confidence intervals over multiple training seeds and a $\log y$-scale for readability where appropriate.
>
> We added percentile confidence intervals in Figure 6 (c) and (d) (p13) using the bootstrap method. This approach captures the variability across measurements while keeping the computational cost of FID manageable, as detailed in Appendix A page 21. We agree that, displaying uncertainty here is essential since the goal is to illustrate a precise trend.
>
> > Appendix E.2, Figure 7: The uncertainty band dominates the plot, making trends hard to interpret. The authors should consider adjusting scale, adding more runs, or plotting relative changes to improve readability. As is, the figure bears limited utility.
>
> We agree that the original plot was not clearly informative. We have therefore revised it in the revision (p36, Figure 9). The new plot is generated with more runs and follows the same methodology as for the FID analyses (for $K = 1, 2, 5$), but where each NLL is computed three times over the entire validation dataset to ensure stability. We now observe a very marginal decrease as $K$ increases, which is consistent with the behavior of $L_{\mathrm{DDSM}}$, given that the likelihood is directly related to the score.
>
> **Clarification and organisation of theoretical claims**
>
> > They cite Xu et al. (2024) noting convergence to similar solutions in diffusion models sharing the same architecture, contradicting their claim p.5: “we could expect models trained on different seeds to converge to noticeably different solutions”.
>
> We agree that the initial phrasing was ambiguous and have clarified it in Section 3.1.1 (p. 4–5). Our intent was to note that while stochasticity in diffusion training (noise sampling and random timesteps) could *be expected to* yield different solutions across seeds, Xu et al. (2024) point out that diffusion models actually converge to similar local minima. The statement about diversity “naturally arising from stochasticity” referred to DE in **supervised learning**, where such stochasticity is known to induce diversity, not to diffusion models.
>
> > This appendix also includes remarks on concept-conditional modeling and linear inverse problems. The authors should consider retitling or splitting into subsections so readers can locate these topics easily.
>
> We agree with your suggestion and added a subsection C.5 p29 accordingly to clearly distinguish the main proposition C.2 and its implications.
>
> > Also, it is unclear why linear inverse problems are mentioned. The section should either be more clearly involved in the manuscript or removed.
>
> We clarified the mention of linear inverse problems in Section 4.2.2 (p. 8) and kept the details in Appendix C.5. This case likewise involves a conditional distribution with a perturbed $\mathbf{x}$, where the likelihood decomposition enables summing scores at each diffusion step.
>
> > Appendix C.4: The proof would read more smoothly with proper declaration of fixed quantities and a lemma-first structure for intermediate steps.
>
> We agree that the original formulation lacked precision, we revised the proof of the proposition to more explicitly state the fixed quantities accordingly to your remark (p28). We find it clearer to present the intermediate algebraic results in the next section, as they are essential for rigor but standard and not required to the proof’s understanding.

---

> ### Author Response · Authors · 2025-10-29
> **Answer to reviewer 9hb2 (3/3)**
>
> **On general methodology**
>
> > Please clarify whether any MCMC/Feynman–Kac correctors were used in experiments ; if not, the scope of the contribution should be specified.
>
> No MCMC or Feynman–Kac correctors were used, as clarified in Appendix C.5 p30. We omitted them to keep the method simple, given that guidance-based approaches already perform very well without such corrections. Additionaly correctors mainly enforce theoretical rigor and bring samples closer to the target distribution, but in our framework this would only align sampling with the PoE formulation, not necessarily improve image quality.
>
> **On the scope of the paper**
>
> Regarding the scope, purpose and framing of our work, we tried to clarify it in the revision, and wanted to elaborate below on the intent and rationale behind our choices.
>
> > Empirical evaluation is dominated by image datasets and score-averaging baselines of Deep Ensemble (DE) models with the same architecture. Two tabular experiments use Random Forest-based scores, without intersecting with deep ensembling, despite different behaviors being observed (no gain in the DE ensembling, while RF shows Domination ensembling gains). As a result, conclusions are mostly retained for unconditional image generation with specific DE methods using score aggregation. The scope of the contribution needs to be backed-up by further experiments, or narrowed down to a precise set of practices.
>
> The focus on image datasets is intentional, as they represent the primary and most established application domain of diffusion models. All deep models used are among the most recognized diffusion architectures, originally trained for image generation tasks. As highlighted in the recent survey [1, Section 3], most major diffusion methods have been developed for image synthesis. Although this reference is a preprint, we included it to provide a current and comprehensive overview of the field.
>
> We deliberately limited our analysis to the simplest possible ensembling setup, since to our knowledge ensembling in diffusion models has not been systematically explored before. We therefore consider uniform combinations of score regressors. All our experiments, including those using RFs, follow this consistent framework, avoiding ensembles with heterogeneous structures or learned weighting schemes. Experiments involving both Deep Ensembles and Random Forests are common in the literature, and we added corresponding clarifications and citations in Section 5 (p. 8).
>
> > Heterogeneous ensembles like mixed architectures (U-Net, CNN, etc.) are not explored
>
> We appreciate this insightful comment. Investigating heterogeneous ensembles with different architectures would actually be an interesting direction. In this work, however, we focus on democratic ensembles, as we emphasize in the paper (Section 3.2 p5; Section 5 p8 & Conclusion p13), in order to study the effect of simple score models aggregation as one could do in a straightforward manner (since it is the first paper on diffusion models ensemble).
>
> [1] Md Manjurul Ahsan, Shivakumar Raman, Yingtao Liu, and Zahed Siddique. A comprehensive survey on
> diffusion models and their applications. arXiv preprint arXiv:2408.10207, 2024
>
> **On references**
>
> > Cited work: Several citations are preprints.
>
> Thank you for this remark, which allowed us to notice that some references were still cited as preprints even though published versions are now available. We have carefully reviewed and updated the bibliography accordingly.
>
> Initially, the manuscript contained nine preprints. Several of them have now been replaced with their official published versions:
> * Albergo et al. (JMLR 2025),
> * Mattei & Garreau (JMLR 2025),
> * Chidambaram et al. (NeurIPS 2024),
> * Havasi et al. (ICLR 2021).
>
> After these updates, four preprints remain:
>
> * Ho & Salimans (2022) and Balaji et al. (2022) — both are highly influential papers (≈990 and 5700 citations, respectively) that are widely referenced across the DM literature.
> * Fort et al. (2019) — a popular work (≈800 citations) that remains a main reference for Deep Ensemble.
> * McAllister et al. (2025) is cited in the conclusion alongside another work as an example for model aggregation.
> * Xu et al. (2024) is an important reference related to the study of diversity in diffusion models.

---

> > ### Comment · Reviewer_9hb2 · 2025-11-02
> > **Review of Rebuttal 1**
> >
> > #### **Overall comment**
> >
> > I thank the authors for incorporating most of my suggested revisions and answering my questions. In particular, the added statistical analyses across experiments make the empirical claims more convincing, and the expanded protocol in Appendix A and the more nuanced conclusions improve the manuscript’s overall soundness. A few core claims still read stronger than the current evidence supports, and several presentation/clarity items would further improve readability from my perspective.
> >
> > #### **Comments and questions**
> > - **(Abstract) "while ensembling the scores generally improves the score-matching loss and model likelihood, it fails to consistently enhance perceptual metrics such as FID."**: This claim remains overly broad. The results indicate that the FID/KID observation holds for image datasets, but specific aggregation rules improve Wasserstein distance on tabular data. The authors are invited to scope this claim to image datasets and soften it for tabular settings to match the evidence. In addition, the Conclusion now contains appropriate nuance and caveats that are missing from the Abstract, thus establishing that nuance at both places would strengthen the paper.
> > - **Code availability**: The repository link is provided, but there are no installation or reproduction instructions, still hindering proper verification and reproduction.
> > - **"(p.5) Xu et al. (2024) showed that diffusion models with different weight initializations (or architectures) tend to converge to similar local minima"**: This wording is confusing as architectural changes alter the parameter space. Two models with the same architecture can differ in weights, but “same local minimum across architectures” is ill-posed. The authors are encouraged to find another formulation that clarifies their statement.
> > - **(p.12) Underfitting or estimation bias?**: I thank the authors for clarifying their claim on RF "underfitting". However, I am still unsure the wording is appropriate. It appears from Figure 11 that Random Forest Arithmetic mean estimator do underestimate the noise, which may not be directly related to an underfitting effect but could arise from an estimation bias.
> > The authors are invited to clarify this possible confusion between statistical bias and capacity effects.
> > - **(p.30) About MCMC/Feynman-Kac correctors: "No such correctors are applied in our experiments, as we focus on the baseline ensembling behavior without additional guidance refinements" and the usage of same-architecture democratic DE**: Although the authors clarified their intent during the rebuttal and at places in the manuscript, these limitations of the study need to be stated in the Discussion/Conclusion to provide further nuance to their claims.
> >
> > #### **Minor issues**
> > - Caption of Figure 3 remains quite uninformative: Figure 3a, 3b, 3c, 3d show filled-up regions that are not described. Points for $K=1, K=2$ seem to have been bootstrapped, but apparently not for $K=4$?
> > - Figure 4 font, size and style do not match that of other figures. Please harmonize for consistency.
> > - Figure 5b now displays a standard deviation for $K=1 \pm 3 \times std$. Please justify the choice of $\pm 3$ standard deviation, or use the classical $\pm 1 std$. Besides, $K=2$ does not have its own standard deviation displayed, which still does not allow to give a hint at whether $K=2$ is statistically significantly worse than $K=1$. Also, the figure is quite small and reading the graph is very challenging.
> > - The authors are invited to use a log-scale for Figure 7a to improve readability.
> > - p.21 show unresolved citation/cross-reference that needs to be patched.
> > - Some formulations would need phrasing improvements. Notably p.21: "taking its 2.5th and 97.5th percentiles for a 95% interval, which is what we do."
> > - p.24 Eq. (23) would benefit from parenthesizing in the sum to avoid confusion.
> > - Missing spacing p.41 "We show in Figure 4that".

---

> > > ### Author Response · Authors · 2025-11-05
> > > **Answer to reviewer 9hb2 (1/2)**
> > >
> > > Thank you for your constructive feedback. We are currently preparing a global update of the manuscript, which will be submitted by Friday, once all reviewer have had time to see the current version and their responses have been received. In the meantime, we provide below the verbatim text of the planned revisions to clearly illustrate the intended modifications. The updated implementation and code repository will also be released by Friday.
> > >
> > > > (Abstract) "while ensembling the scores generally improves the score-matching loss and model likelihood, it fails to consistently enhance perceptual metrics such as FID.": This claim remains overly broad. The authors are invited to scope this claim to image datasets and soften it for tabular settings to match the evidence.
> > >
> > > We changed the abstract in the first revision to “ensembling the scores” to exclude the MoE in the message. Still, we agree that the distinction between image and tabular datasets was unclear. We plan to update the abstract as follows:
> > >
> > > "We find that while ensembling the scores generally improves the score-matching loss and model likelihood, it fails to consistently enhance perceptual quality metrics such as
> > > FID **on image datasets**. We confirm this observation across a breadth of aggregation rules using Deep Ensembles, Monte Carlo Dropout, ~~and Random Forests~~ on CIFAR-10, FFHQ, ~~and tabular data~~. We attempt to explain this discrepancy by investigating possible explanations, such as the link between score estimation and image quality. **We also look into tabular data through random forests, and find that one aggregation strategy outperforms the others.** "
> > >
> > > > "(p.5) Xu et al. (2024) showed that diffusion models with different weight initializations (or architectures) tend to converge to similar local minima": This wording is confusing as architectural changes alter the parameter space.
> > >
> > > We agree that the wording “similar local minima” can be misleading, especially when architectures differ and thus define distinct parameter spaces. Our intended meaning was that diffusion models often converge toward similar learned transformations. We plan to update the sentence as follows:
> > >
> > > "Xu et al. (2024) showed that diffusion models with different weight initializations (or architectures) tend to converge to similar ~~local minima~~ **functions**"
> > >
> > > > (p.12) Underfitting or estimation bias?: It appears from Figure 11 that Random Forest Arithmetic mean estimator do underestimate the noise, which may not be directly related to an underfitting effect but could arise from an estimation bias. The authors are invited to clarify this possible confusion between statistical bias and capacity effects.
> > >
> > > We agree that underfitting was not the appropriate term in this context, and that the issue is better described as an *underestimation bias*. Thus we plan to modify the underfitting mentions as follows:
> > >
> > > "(Contributions p.2) We highlight two settings in which ensembling can be helpful. First, the Mixture of Experts provides small benefits when the models are sufficiently complementary, allowing their diversity to be effectively exploited. Second, in the case of decision-tree aggregation, one method alleviates an ~~underlying underfitting~~ **underestimation bias** issue, leading to improved performance."
> > >
> > > "(Section 5.2.4 p.12) This noise is in fact systematically underestimated during generation, as shown in Figure 11. Arithmetic mean exhibits a lower overall standard deviation for all $t > 0$, supporting ~~the idea that~~ **the hypothesis of a statistical bias, namely that** it fails to capture the true noise magnitude. ~~It may be caused by an underfitting effect coming from decision trees.~~"
> > >
> > > > (p.30) About MCMC/Feynman-Kac correctors: limitations of the study need to be stated in the Discussion/Conclusion to provide further nuance to their claims.
> > >
> > > We will add a paragraph in the conclusion (p.13, second-to-last paragraph) to clarify our intent as follows:
> > >
> > > "We showed that the mean of the scores at each step does not sample the PoE. While Feynman–Kac (Skreta et al., 2024) or MCMC-based (Du et al., 2023) correctors  could in principle adjust the sampling process to better target the PoE, we deliberately omitted them to restrict the work to baseline ensembling. Moreover, guidance-based approaches already perform well without the additional theoretical rigor introduced by such corrections. Nonetheless, future work could further explore this direction."

---

> > > ### Author Response · Authors · 2025-11-05
> > > **Answer to reviewer 9hb2 (2/2)**
> > >
> > > **Minor issues**
> > >
> > > > Caption of Figure 3 remains quite uninformative: Figure 3a, 3b, 3c, 3d show filled-up regions that are not described. Points for $K = 1$, $K = 2$ seem to have been bootstrapped, but apparently not for $K = 4$?
> > >
> > > We refer to the details in Appendix A on the caption and on page 8. For a given maximum ensemble size $K_{\text{max}}$ (which is the total number of trained models), we report confidence intervals for the variability across model subsets when evaluating FID for smaller ensemble sizes (e.g. for $K=2$ we take all the subsets of size 2 in the whole set of size $K$ and compute the mean and stds of FIDs, for $K=1$ we take all the individual models and compute mean and stds of FIDs). For $K=5$ there is only one combination (1,2,3,4,5) if $K_{\text{max}} = 5$ (e.g. on CIFAR-10), so there is a single run given that each FID is computed one time. In FFHQ we have $K_{\text{max}}= 4$, and thus we similarly have one run for $K=4$.
> > >
> > > However, if one wished indeed to estimate the variability induced by the metric itself (e.g., FID) rather than by model combinations, this could be done by computing the metric multiple times. Since this is computationally expensive (each FID requires 10 k generated samples), we instead apply a bootstrap procedure to approximate this variability. The bootstrap method is only used for FID in Figure 7b, where one fixed model is used and we need to measure a precise trend of the FID. Note that the resulting variability is negligible compared to the model intialization induced variability. Details are in Appenix A (Section A.3.2, page 21).
> > >
> > > > Figure 5b now displays a standard deviation for $K = 1 \pm 3 \times \textit{std}$. Please justify the choice of $\pm 3$ standard deviation, or use the classical $\pm 1\textit{std}$.
> > >
> > > We realized that the standard deviation is not very meaningful to represent the uncertainty for $K_{\text{max}} = 2$, so we plan to display instead the minimum and maximum values in the revision.
> > >
> > > >   Besides, $K = 2$ does not have its own standard deviation displayed, which still does not allow to give a hint at whether $K = 2$ is statistically significantly worse than $K = 1$.
> > > > p.21 show unresolved citation/cross-reference that needs to be patched.
> > >
> > > Given the explanations given above, since we only saved the intermediate checkpoints of 2 models we have $K_{\text{max}} = 2$. Thus the procedure of variability calculation is the same reasoning as for $K_{\text{max}} = 5$, as suggested in the Appendix p.21 in which we reference Figure 5b (which corresponded to the unresolved citation).
> > >
> > > > Some formulations would need phrasing improvements. Notably p.21: "taking its 2.5th and 97.5th percentiles for a 95% interval, which is what we do."
> > >
> > > We modify the sentence as follows:
> > >
> > > "The confidence interval is then obtained from this empirical distribution, either using its standard deviation or, as in our case, by taking the 2.5th and 97.5th percentiles to form a 95% interval."
> > >
> > > Thank you for your other remarks; they will be taken into account in the next revision.

---

### Author Response · Authors · 2025-10-29
**General comment**

We thank all reviewers for their valuable and detailed feedback.

Reviewers 9hb2 and 22sG responded “Yes” to the evaluation item regarding the potential interest of our findings to TMLR’s audience. In particular the reviewers agreed that our work “bears interest for the ML community in general, both from theoretical and practical perspective” (Reviewer 9hb2) and appreciated that “the community, especially practitioners, would be interested to see how useful (or not) the ensembling techniques are for diffusion models” (Reviewer wo7R).

Reviewers wo7R and 22sG selected "Yes" for whether the claims made in the submission are supported by accurate, convincing, and clear evidence. We are glad that our theoretical results were found insightful, as they “provide insights as to why averaging scores reduces the DDSM objective and motivate why ensembling for score estimation is challenging” (Reviewer 9hb2). We also appreciate the remark that our proof “is a valuable and elegant theoretical contribution” clarifying that diffusion and Product-of-Experts “do not commute” (Reviewer 22sG). Finally, we thank the reviewers for their positive assessment of our empirical results, noting that “the core findings on score loss are accurate and clearly shown” (Reviewer 9hb2), that “experiments on different datasets and models demonstrate the effect of ensembling on model performance” (Reviewer wo7R), and that our main claim “is strongly supported” (Reviewer 22sG).

We thank again the reviewers for their detailed reviews which will help improve the paper. Following the reviewer's remarks, we have made the following modifications to the paper **(modifications are in red in the PDF)**:
* (i) As suggested by Reviewer 9hb2 and Reviewer 22sG, we revised the overall message of the paper to better reflect the scope of our findings and to highlight the two cases where ensembling did provide a gain in the introduction, which we agree are important to emphasize.
* (ii) As suggested by Reviewer 9hb2 we clarified all points that were unclear regarding the confidence intervals (standard deviations) specifying under which conditions they are reported, and how they are computed. We also computed confidence bounds in Figure 7 (c) and (d) in page 12.
* (iii) As suggested by Reviewer 9hb2 we have published the code of the paper and we provided a link to an anonymous repository in the abstract of the paper (Our Python code is available at (https://anonymous.4open.science/r/score_diffusion_ensemble-B758).
* (iv) As suggested by Reviewer wo7R, we added an experiment on the effect of the number of diffusion steps during sampling on ensembling results (Section 5.1 page 9; and Figure 10 in appendix, page 35), which we agree is an important complementary analysis.

There are also minor changes on the paper that are not direct suggestions of reviewers. We corrected Figure 3 (a,b), as one mean point ($K=2$) of each curve was miscomputed. We also introduced a slightly stronger Proposition 4.1 on monotonicity, which strengthens the theoretical insight.

---

> ### Author Response · Authors · 2025-11-07
> **General comment 2**
>
> As announced in my response to reviewer 9hb2, the paper has been updated today for the second revision, incorporating the modifications discussed in our exchange. The new updates are highlighted in orange, while the previous updates in red have been kept unless they were replaced by a newer orange modification.
>
> The accompanying repository has been updated with run instructions, and includes the five CIFAR-10 and four FFHQ checkpoints used in our experiments.
>
> We thank again all reviewers for their time and feedback.

---

### Decision · Action_Editor_u1ic · 2026-01-05

**Recommendation:** Accept as is

**Additional Comments:**

While reviewers acknowledged the relevance of the empirical evaluation, they raised concerns regarding the clarity of  the contribution and the scope of the paper. They also had several questions and comments  on the experimental results (limited perceptual gains, statistical significance). The authors revised the paper and reorganized parts of the paper contributions and empirical results. This better emphasizes the main claims, and clarifies the methodology and technical results (settings in which ensembling provides measurable benefits, confidence intervals). The fact that the authors  released the code, and added new experiments improves the contribution significantly.

**Audience:**

Yes

**Audience Explanation:**

Diffusion models have become go-to generative models for many applications and a very active reserach activity provides new sampling strategies and theoretical guarantees for such approaches.

Providing a quantitative study to analyze the impact of ensembling for these methods either using standard U-nets or random forests is of interest for a large ML community.

The poor improvements in sample quality, even when the score matching loss is improved, is interesting for practitioners.

**Claims And Evidence:**

Yes

**Claims Explanation:**

This paper investigates the impact of ensembling strategies in diffusion models and provides an empirical study using multiple methodologies and several datasets. The main contribution of the paper and the main claim is that ensembling generally yields only marginal improvements in sample quality, even when the score matching loss is improved.

The authors claims are supported with different ensembling approaches obtained either by training independent U-Net architectures or by learning distinct Random Forest estimators of the score function at each noise level. The performance is assessed using standard perceptual metrics, including in particular FID and KID on CIFAR-10 and FFHQ.

In addition, the authors tested a large variety of aggregation rules to assess their claims.